# *P. aeruginosa* CtpA protease adopts a novel activation mechanism to initiate the proteolytic process

Hao-Chi Hsu[1], Michelle Wang [ID]2, Amanda Kovach [ID]1, Andrew J Darwin [ID]2✉ & Huilin Li [ID]1✉

## Abstract

During bacterial cell growth, hydrolases cleave peptide cross-links between strands of the peptidoglycan sacculus to allow new strand insertion. The *Pseudomonas aeruginosa* carboxyl-terminal processing protease (CTP) CtpA regulates some of these hydrolases by degrading them. CtpA assembles as an inactive hexamer composed of a trimer-of-dimers, but its lipoprotein binding partner LbcA activates CtpA by an unknown mechanism. Here, we report the cryo-EM structures of the CtpA–LbcA complex. LbcA has an N-terminal adaptor domain that binds to CtpA, and a C-terminal superhelical tetratricopeptide repeat domain. One LbcA molecule attaches to each of the three vertices of a CtpA hexamer. LbcA triggers relocation of the CtpA PDZ domain, remodeling of the substrate binding pocket, and realignment of the catalytic residues. Surprisingly, only one CtpA molecule in a CtpA dimer is activated upon LbcA binding. Also, a long loop from one CtpA dimer inserts into a neighboring dimer to facilitate the proteolytic activity. This work has revealed an activation mechanism for a bacterial CTP that is strikingly different from other CTPs that have been characterized structurally.

**Keywords** Protease Activation; *Pseudomonas aeruginosa*; Cell Wall; Peptidoglycan Hydrolases; Protein Complex
**Subject Categories** Microbiology, Virology & Host Pathogen Interaction; Post-translational Modifications & Proteolysis; Structural Biology

## Introduction

Peptidoglycan is an essential component of almost all bacterial cell envelopes (Vollmer et al, 2008). It forms a mesh-like sacculus to maintain cell integrity by helping to resist turgor pressure. Parallel linear glycan strands made up of alternating *N*-acetylglucosamine (Glc*N*Ac) and *N*-acetylmuramic acid (Mur*N*Ac) are crosslinked together via peptide bonds between short peptides that are attached to each Mur*N*Ac molecule. Mammalian cells lack peptidoglycan, which makes it an excellent target for selective antibacterial agents. Cell growth requires enlargement of the peptidoglycan sacculus. Two classes of enzymes, hydrolases and synthases, work coordinately to ensure the insertion of new peptidoglycan strands into the mesh (Höltje, 1998; van Heijenoort, 2011). The cross-link hydrolases MepS, MepM, and MepH were identified as playing a role in *Escherichia coli* cell elongation (Singh et al, 2012). However, the activity of cell wall hydrolases must be tightly regulated to prevent sacculus rupture and cell death. In *E. coli*, the carboxyl-terminal processing protease (CTP) Prc was found to work with the outer membrane lipoprotein adaptor NlpI to modulate the level of MepS by proteolysis (Singh et al, 2015). More recently, Prc was also shown to degrade MepM and MepH as well (Jeon and Cho, 2022; Kim et al, 2021).

Bacterial CTPs are designated as family S41A in the MEROPS peptidase library (Rawlings et al, 2002). Their name arises from early observations that they can cleave their substrates close to the C-terminus (Anbudurai et al, 1994; Hara et al, 1991; Silber et al, 1992). However, it is now known that at least one CTP cleaves the N-terminal region of its substrate (Deng et al, 2018). Bacterial CTPs are located outside of the cytoplasm and are ATP-independent. Their active site contains a conserved Ser–Lys catalytic dyad (Keiler and Sauer, 1995). All CTPs also contain a PDZ domain, which plays an important role in substrate recognition and regulation of protease activity (Beebe et al, 2000; Sommerfield and Darwin, 2022; Wilken et al, 2004). CTPs participate in many bacterial processes, including cell growth, virulence, DNA damage sensitivity, and sporulation (Burby et al, 2018; Campo and Rudner, 2007; Chakraborty and Darwin, 2021; Mastny et al, 2013; Seo and Darwin, 2013; Sommerfield and Darwin, 2022; Srivastava et al, 2018). They also play a role in regulating cell wall cross-link hydrolases by degrading them in both *E. coli* and *Pseudomonas aeruginosa* (Singh et al, 2015; Srivastava et al, 2018).

The activation mechanism of some CTPs has been studied. In the *E. coli* Prc–NlpI system, the outer membrane-anchored lipoprotein NlpI has four tetratricopeptide repeats (TPR) and forms a homodimer with the TPRs located on the outside edges of the dimer (Su et al, 2017). Two separate Prc molecules bind to the NlpI homodimer, by interacting primarily with TPR2, to form a linear Prc–NlpI–NlpI–Prc complex (Su et al, 2017). Prc is a monomeric bowl-shaped structure with a lid-like PDZ domain. After sensing the substrate by its interaction with the Prc PDZ domain, the active site is remodeled, with the catalytic residues realigned into their active state (Chueh et al, 2019). Another example is provided by *Bacillus subtilis* CtpB, which is involved in a regulated intramembrane proteolysis pathway that cleaves the SpoIVFA regulator to trigger the late stages of sporulation (Zhou

[1]Department of Structural Biology, Van Andel Institute, Grand Rapids, MI, USA. [2]Department of Microbiology, New York University Grossman School of Medicine, New York, NY, USA. ✉E-mail: andrew.darwin@med.nyu.edu; Huilin.Li@vai.org

and Kroos, 2005). CtpB forms a dimeric ring-like scaffold with the proteolytic site of each subunit guarded by the PDZ domain. After binding to the C-terminus of its substrate, the PDZ domain shifts its position to allow substrate access to the active site and realignment of catalytic residues (Mastny et al, 2013).

*Pseudomonas aeruginosa* is one of the most common nosocomial pathogens, with both intrinsic drug resistance and emerging multidrug-resistant strains worldwide (Crivaro et al, 2009; Moore and Flaws, 2011; Paterson, 2006; Tumbarello et al, 2013). In contrast to *E. coli*, *P. aeruginosa* has two CTPs. One is an apparent ortholog of *E. coli* Prc that is known as Prc or AlgO, and the other is CtpA. However, in contrast to *E. coli*, it is CtpA rather than Prc that forms a complex with an outer membrane lipoprotein adaptor to degrade cell wall cross-link hydrolases in *P. aeruginosa* (Srivastava et al, 2018; Su et al, 2017). The macromolecular structure of CtpA is very different to Prc. CtpA forms a trimer-of-dimers triangular-shaped hexamer (Hsu et al, 2022). The folding of an individual CtpA monomer resembles that of a *B. subtilis* CtpB monomer. Each CtpA monomer comprises one N-terminal dimerization region (NDR), one PDZ domain, one core domain, and one C-terminal dimerization region (CDR). Two CtpA molecules form a dimer via NDR interactions, but unlike CtpB, three of these dimers link together by their CDRs to form the triangular-shaped hexamer (Hsu et al, 2022). In this structure, the active site of CtpA is blocked by the PDZ domain, and the catalytic residues are in an inactive configuration, which is consistent with CtpA requiring the adaptor lipoprotein LbcA for activity (Srivastava et al, 2018). LbcA is a monomer consisting entirely of α-helices, which form a long N-terminal extension and a C-terminal superhelical TPR domain (Hsu et al, 2022). Deletion of the first N-terminal helix causes the same phenotype as *lbcA* and *ctpA* null mutations, suggesting that the N-terminal helix of LbcA is critical for CtpA activation (Hsu et al, 2022). The C-terminal superhelical TPR domain of LbcA, which is formed by 11 TPRs, might be involved in substrate binding and delivery to CtpA (Blatch and Lassle, 1999; Zeytuni and Zarivach, 2012). Five proteins have been identified as substrates of the CtpA–LbcA complex. Four are predicted to be peptidoglycan cross-link hydrolases: the LytM/M23 family peptidases MepM and PA4404 and the NlpC/P60 family peptidases PA1198 and PA1199 (Srivastava et al, 2018). The fifth substrate is a predicted outer membrane lipoprotein of unknown function, which has a C-terminal domain predicted to bind to peptidoglycan noncovalently (Chakraborty and Darwin, 2021).

There are remarkable differences between the structural arrangements of *P. aeruginosa* CtpA and *E. coli* Prc, and their LbcA and NlpI adaptor proteins are not homologous. Therefore, we hypothesized that the way in which LbcA interacts with and activates CtpA might be unique. To gain insight into this, we have used cryo-EM single particle analysis to determine the structure of the CtpA–LbcA complex. This has revealed a protease activation model that is distinct, not only from Prc, but also from all previously studied bacterial CTPs.

# Results

## Cryo-EM of the CtpA–LbcA protease complex

To reduce the complexity caused by unstructured regions, in our expression constructs, we removed the regions encoding the

N-terminal 37 amino acids of CtpA consisting of the type I signal sequence and a bacterial low complexity region, and the N-terminal 31 amino acids of LbcA including the type II signal sequence and lipidation site (Cys-17) for anchoring to the outer membrane (Hsu et al, 2022) (Fig. 1A; Appendix Table S1). We also found that LbcA was slowly degraded when mixed with the wild-type CtpA. Therefore, we used the previously described catalytically inactive CtpA(S302A) mutant in this study (Seo and Darwin, 2013; Srivastava et al, 2018). Co-expressed CtpA and LbcA were purified as a complex with an apparent molar ratio of ~2:1 as judged by band intensity in the Coomassie Blue (Fig. 1B) and SYPRO Ruby (Appendix Fig. S1) stained SDS-PAGE gels.

We derived by homogenous refinement a cryo-EM map of the CtpA–LbcA complex at an overall resolution of 3.84 Å (Appendix Tables S2 and 3). The map retains the overall triangular shape of the trimer-of-dimers CtpA-alone hexamer but with partial LbcA densities at each vertex (Fig. 1C; Appendix Figs. S2 and 3). The top vertex has a better-defined LbcA supercoil density, whereas the two lower vertices have much weaker LbcA densities, probably due to their flexibility. However, a full LbcA supercoiled TPR domain is visible at the top vertex when the map is rendered at a low display threshold (Appendix Fig. S4a). We next performed a heterogeneous refinement and obtained four 3D EM maps (Fig. 1D). We found in all these maps each vertex is occupied by an LbcA. However, LbcA can bind to either the front or the back of the CtpA dimer at each vertex. We observed two LbcA binding patterns: all three LbcA molecules bound on the same side (front) of the CtpA hexamer, or two LbcA on the front side and one on the back. The fact that two different binding modes were observed suggests that three LbcA molecules bind the three individual CtpA dimers independently, and that there is no LbcA binding cooperativity at the three vertices.

We next carried out a local refinement focusing on the upper one-third region of the complex containing a CtpA dimer and the best LbcA density, which obtained an EM map at an average resolution of 3.55 Å (Appendix Figs. S2 and S5). In this map, the CtpA dimer and its interface with LbcA have a resolution better than 3.3 Å (Fig. 1E; Appendix Fig. S5), enabling us to build an atomic model by referencing to the published crystal structures of the CtpA hexamer (PDB ID 7RQH) (Hsu et al, 2022) and LbcA (PDB ID 7RQF) (Hsu et al, 2022) (Fig. 1F). The structure comprises two CtpA monomers, a substrate peptide, and the N-terminal region of LbcA. In the atomic model, as described below, LbcA approaches the CtpA dimer from the front and is ordered in the N-terminal adaptor domain (Fig. 1F). Interestingly, only the front CtpA is in an activated configuration while the back CtpA remains inactive. We will refer to the inactive protease as CtpA' from now on.

## The LbcA N-terminus refolds into an adaptor domain to interact with CtpA

In the crystal structure of LbcA alone, the N-terminal four α-helices (H1c-H4c) preceding the hinge region (H5c-H6c) form a long extension (Hsu et al, 2022) (Fig. 2A,B). We found that this extension (H1c-H4c) refolds into an adaptor domain of six shorter antiparallel α-helices (H1–H6) upon interaction with CtpA, so that the two hinge-domain α-helices (H5c-H6c) now become H7–H8 (Fig. 2A,C). It is likely that the LbcA N-terminus (aa 1–163) is

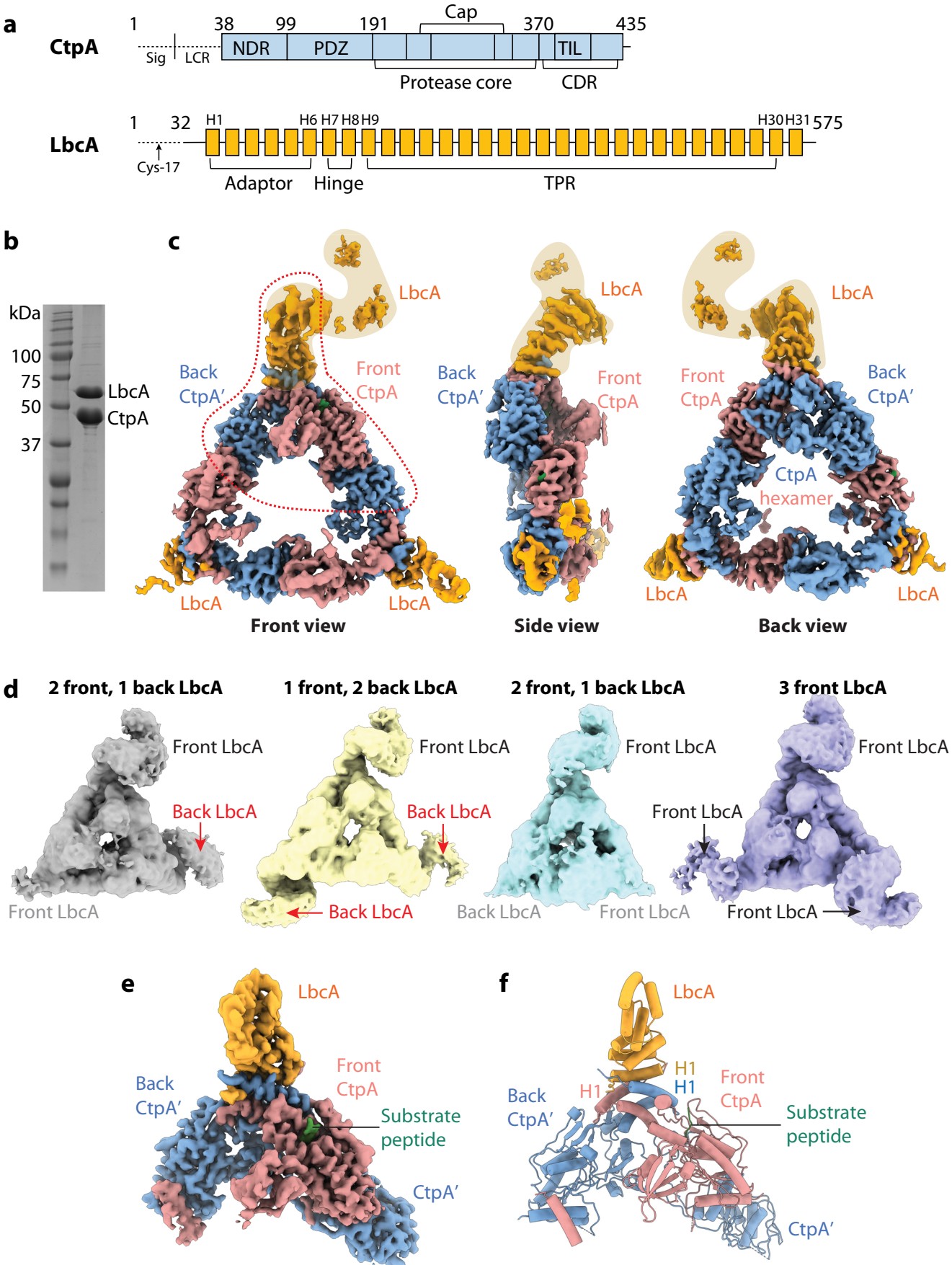

 **Figure 1. Cryo-EM of the CtpA–LbcA complex.**

(A) Domain architectures of CtpA and LbcA. (B) SDS-PAGE gel of the purified CtpA–LbcA complex suggesting a 2:1 molar ratio of CtpA and LbcA. (C) Overall EM map of the CtpA–LbcA complex in three orthogonal views—front, side, and back. The dashed red shape marks the region where local refinement was performed to obtain the EM map shown below. (D) Four EM classes in front view revealing that LbcA can bind independently from either front or back to the three CtpA hexamer vertices. Note that LbcA molecules labeled in gray are invisible at the display level but become visible at a lower threshold. (E) Front view of the locally refined EM map consisting of one CtpA dimer, one LbcA adaptor domain, and some residual densities at the CtpA dimer–dimer interface. (F) Front view of the locally refined CtpA–LbcA structure. Source data are available online for this figure.

disordered in solution and the four-helix extension in the crystal structure is induced by crystal packing, and that the N-terminal peptide folds into the adaptor domain when it interacts with CtpA, as observed here in the CtpA–LbcA complex.

CtpA dimerization is mediated by a four-helix bundle formed by the N-terminal α-helices (H1 and H2) of two CtpA monomers, as observed previously in the CtpA hexamer crystal structure (Hsu et al, 2022). We found that the LbcA N-terminal α-helix 1 (H1) inserts into a space between the two H1 of the CtpA dimer (Fig. 2D). The space between the two CtpA H1 is widened by 8 Å to accommodate the LbcA H1 (Fig. 2E). The CtpA–LbcA interface is extensive but contains only three hydrogen bonds: LbcA Phe-48 forms one H-bond with Lys-57 and another H-bond with Asp-66 of CtpA, and LbcA Gly-46 forms the third H-bond with CtpA' Pro-40 (Fig. 2F). The CtpA–LbcA interface is dominated by hydrophobic interactions (Fig. 2D,G,H). Specifically, a cluster of three hydrophobic residues Leu-53, Leu-57, and Leu-61 of LbcA H1 contacts hydrophobic Leu-46, Phe-49, and Leu-53 of CtpA H1, and another LbcA hydrophobic cluster comprised by Tyr-54, Val-58, and Ala-62 interacts with CtpA Leu-41 and Leu-43 (Fig. 2G). The LbcA H1 forms similar hydrophobic interaction with CtpA' H1, involving LbcA Phe-48, Leu-53, Leu-56, Leu-57, Leu-61, and Val-87 and CtpA' Leu-43, Leu-46, Phe-49, and Leu-53 (Fig. 2H).

## The hydrophobic interface between LbcA and CtpA is essential for function

To investigate the importance of the hydrophobic interface, we selected two hydrophobic residues at the interface in each protein, Leu-46 and Ala-50 of CtpA, and Leu-57 and Val-87 of LbcA, for mutational studies in *P. aeruginosa*. Mutations that maintained the hydrophobicity in CtpA (L46A and A50V) did not markedly affect degradation of the substrate PA1198, whereas mutations to basic residues (L46K and A50K) abolished PA1198 degradation (Fig. 2I). Likewise, the conservative mutations L57A and V87A in LbcA maintained substrate degradation activity, whereas the basic mutations L57K and V87K eliminated the protease activity (Fig. 2I). Similar results were obtained when another CtpA substrate was monitored, PA1048 (Appendix Fig. S6a,b). To monitor the effects of the mutations on CtpA–LbcA interaction, we used a bacterial two-hybrid assay based on reconstitution of a cyclic AMP signaling cascade (Karimova et al, 1998). This probes for the proximity of two fragments (T18 and T25) of the catalytic domain of *Bordetella pertussis* adenylate cyclase (Cya). Different proteins are fused to T18 and T25, restoring Cya activity if they associate. In an *E. coli* Δ*cya* mutant this activates cAMP-CRP-dependent genes such as those required for maltose catabolism, which produces red colonies on MacConkey-maltose agar. This analysis revealed that the conservative hydrophobic substitutions maintained CtpA–LbcA

interaction, whereas basic substitutions that introduced a charge into the interface destroyed it (Fig. 2J). Therefore, the data from these mutational studies support the importance of the hydrophobic interface for CtpA–LbcA interaction and CtpA proteolytic activation in vivo.

## LbcA binding activates the front CtpA protease in the CtpA dimer

The core domains of the two subunits of a CtpA dimer within the CtpA hexameric crystal structure are nearly equivalent (Hsu et al, 2022). Upon LbcA binding, the structures of the front CtpA and the back CtpA' differ dramatically (Fig. 3A). The front CtpA undergoes a series of conformational changes, in particular the PDZ domain is far away from the position in the back CtpA', when the two protease structures are aligned by their respective core domains. Ser-302, Lys-327, and Gln-331 were previously proposed to be the catalytic triad of CtpA, based on the homologous CtpB structure (Hsu et al, 2022; Mastny et al, 2013). In the reported CtpA hexamer structure, which is in the inactive state, the distance between Lys-327 and Gln-331 is 9 Å (Hsu et al, 2022). Interestingly, we found that, in the back CtpA' structure, Ala-302, Lys-327, and Gln-331 are arranged with distances over 6 Å, resembling those of the inactive CtpA hexamer structure (Fig. 3B). However, in the front CtpA, all distances between Ala-302, Lys-327, and Gln-331 are shortened to below 3.3 Å, except for the distance between Ala-302 and Lys-327 that is shortened to only 5.2 Å (Fig. 3C). These observations suggest that the back CtpA' remains inactive upon LbcA binding, whereas the front CtpA has been activated. The larger distance between Ala-302 and Lys-327 (5.2 Å) in the active structure is most likely due to the S302A mutation. The CtpA protease core (aa 191–370) is composed of a cap region and a core domain (Fig. 1A). We found that the assigned inactive CtpA' structure superimposes well with the known inactive structure in the CtpA-alone hexamer (Fig. 3D). However, the active CtpA structure is drastically different, with its NDR, cap, and PDZ having moved by 8 Å, and 44 Å, respectively, compared to those of the inactive CtpA (Fig. 3E).

The CtpA protease substrate binding pocket is encircled by α-helix H3 that connects the NDR and the PDZ domain, the cap, the core domain, with the PDZ domain lining the bottom of the pocket. In the inactive CtpA', the distance between the cap and the core is 32 Å, resulting in a large substrate pocket of 14 Å in diameter (Fig. 3F). In the active CtpA structure, the rearrangement of the NDR has triggered a shift of helix H3 toward the core domain and a relocation of the PDZ domain towards the front (Fig. 3G). In addition, the cap has lowered down by 8 Å to narrow down the substrate pocket to 8.7 Å, perhaps leading to the trapping of an endogenous substrate peptide in the narrowed substrate binding

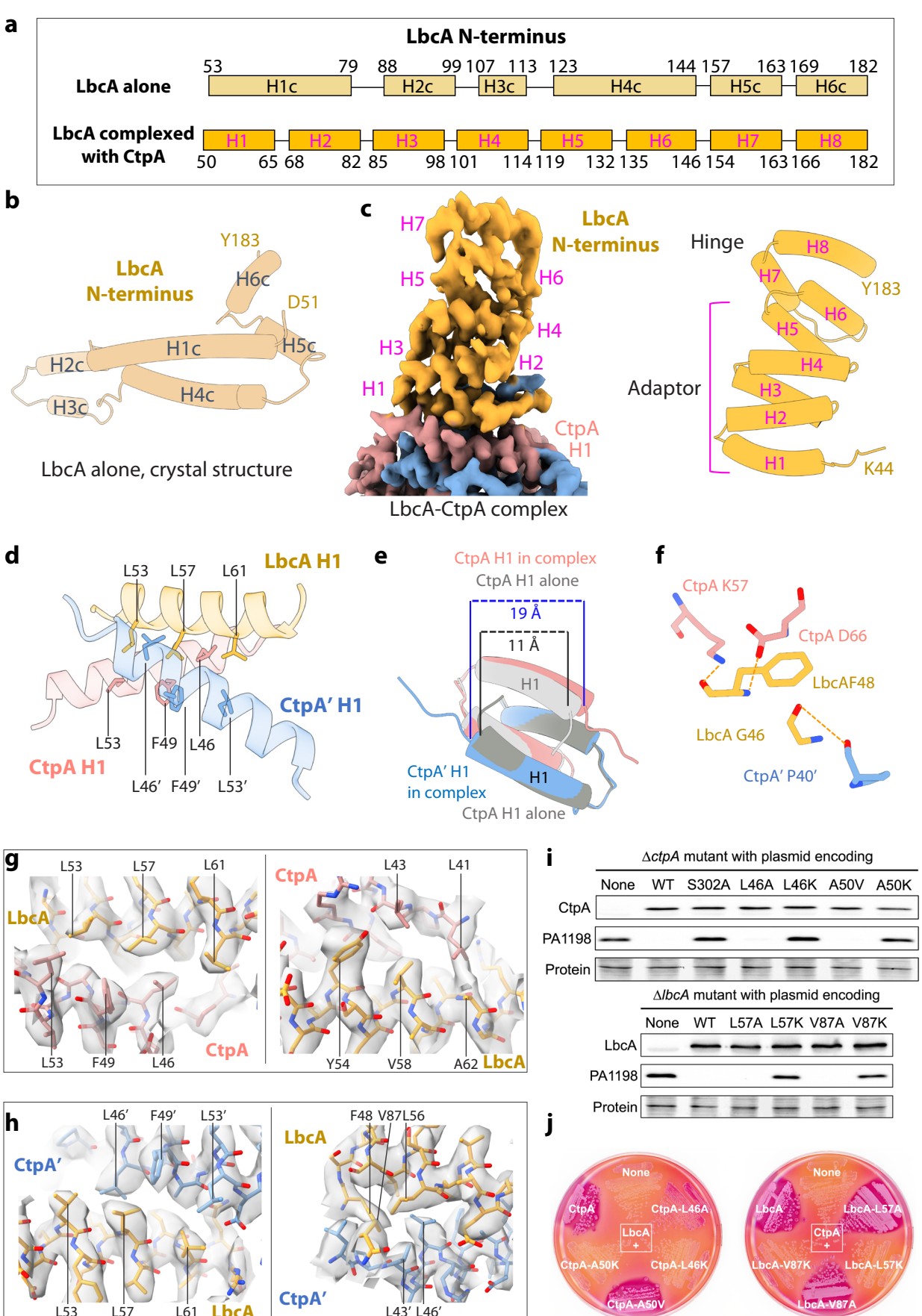

◀ **Figure 2.   The hydrophobic interface between CtpA and LbcA.**

(**A**) Comparison of the LbcA N-terminus in the crystal structure of LbcA alone and in the current EM structure in complex with CtpA, showing refolding of this region. (**B**) The N-terminus of LbcA alone contains six α-helices to form the long extension (PDB ID 7RQF). (**C**) Left: EM map of the refolded N-terminal adaptor domain of LbcA. Right: Cartoon view of the adaptor domain structure of LbcA complex with CtpA. (**D**) The LbcA H1 binds between the two H1 at the NDR of the top CtpA dimer. (**E**) The LbcA H1 binding forces the two CtpA NDR H1 to move apart by 8 Å. (**F**) Three hydrogen bonds between CtpA H1 and LbcA H1. (**G**, **H**) EM map in transparent gray superimposed with the atomic model in sticks showing the hydrophobic interactions between CtpA H1 (pink, **G**) and CtpA′ H1 (blue, **H**) with LbcA H1 (orange). (**I**) Effect of CtpA mutations (top) or LbcA mutations (bottom) on PA1198 substrate degradation in vivo. CtpA, LbcA and PA1198 were detected by immunoblot with polyclonal antisera; loading was monitored by Ponceau S total protein staining of the nitrocellulose membrane used for detection (protein). Data are one representative, but at least three biological replicates of each strain have been analyzed in the laboratory. (**J**) Bacterial two-hybrid analysis of LbcA–CtpA interactions in *E. coli* BTH101 grown on MacConkey-maltose agar. On the left, strains contained a plasmid encoding LbcA fused to the C-terminus of Cya-T18 and a second plasmid encoding CtpA fused to the N-terminus of Cya-T25, derivatives with the indicated CtpA mutations, or Cya-T25 only (None). On the right, strains contained a plasmid encoding CtpA fused to the N-terminus of Cya-T25 and a second plasmid encoding LbcA fused to the C-terminus of Cya-T18, derivatives with the indicated LbcA mutations, or Cya-T18 only (None). Data are one representative of two biological replicates. Source data are available online for this figure.

pocket, as described below. Furthermore, the large PDZ domain shift opens the bottom of the substrate pocket, converting the substrate binding pocket into a substrate binding tunnel with an open exit (Fig. 3G). It is likely that these large movements have led to the rearrangement of the catalytic residues into the active configuration (Fig. 3C).

We next examined the in vivo functional impact of mutating the hypothetical catalytic triad residues (Ser-302, K-327, and Gln-331). The S302A and K327A mutations abolished the proteolytic activity of CtpA in vivo as expected, but a Q331A mutation had only a minor effect (Fig. 3H). As His-84 had been brought within hydrogen bonding distance of Lys-327 in the active CtpA structure (Fig. 3C), we also analyzed a H84A mutant and found that this mutation reduced CtpA activity by a moderate level (Fig. 3H). All of these mutations also compromised the degradation of another CtpA substrate, PA1048, although the Q331A defect was more obvious (Appendix Fig. S6c). Therefore, in the remaining mutagenesis experiments we used PA1198 exclusively as a model substrate, as we have before. The classic catalytic triad of a serine protease is coordinated by a nucleophile, a basic residue, and an acidic residue. The base deprotonates the hydroxyl group of serine to generate the nucleophile, whereas the acid stabilizes the base by restricting the side chain rotation of the base (Matthews et al, 1967). Our mutational analysis of Ser-302 and Lys-327 confirms Ser-302 as the nucleophile and Lys-327 as the base. However, the H84A mutation had a stronger impact on the activity of CtpA than Q331A (Fig. 3H). His-84 may be a better candidate than Gln-331 to function as the acid of a CtpA catalytic triad. The histidine imidazole ring has a p*Ka* around 6 making it capable of both donating and accepting a proton in physiological conditions. However, neither Gln-331 nor His-84 is essential for the activity of CtpA.

## Substrate peptide recognition by the active CtpA

A surprising observation in the CtpA–LbcA complex structure was an extra elongated density in the substrate binding tunnel of the active CtpA that was described above (Figs. 3G and 4A–C). The tunnel reaches the active site Ser-302 at one end and opens toward the TPR domain of LbcA at the other end. The linear density passes through the mutated active site Ala-302 and then vanishes in a nearby space between Pro-245 and Lys-327 (Fig. 4D). We tentatively assigned this density as a substrate peptide that co-purified with the protease and have built a seven-alanine atomic model in the linear density. The heptapeptide is coordinated by

main chain hydrogen bonding with an antiparallel β-strand composed of Pro-245–Gly-246–Gly-247–Val-248–Leu-249 at the left and another antiparallel β-strand composed of Gln-331–Thr-332–Val-333 to the right (Fig. 4D). Such antiparallel β-strand based substrate peptide recognition is also found in the 20 S proteasome core particles (Borissenko and Groll, 2007; Groll and Huber, 2004). Interestingly, the cap of the CtpA core orients Leu-249 and Val-330 toward the P1 and P3 residues, Val-248 and Val-333 surround the P2 residue, and the Phe-325 phenyl ring is close to the P2′ residue. These observations suggest that hydrophobicity may play a role in substrate peptide binding and selectivity. To examine if the precise configuration of the catalytic pocket near the substrate scissile bond is important for the protease activity, we mutated Gly-246 and Phe-325 to methionine and alanine, respectively. Both mutations significantly reduced degradation of the PA1198 substrate in vivo, which was almost as severe as the effect of substitution of the catalytic Ser residue to alanine (S302A) (Fig. 4E). Structural inspection suggests that the large methionine side chain in the G246M mutant may block the substrate P1 binding site thereby breaking the antiparallel substrate peptide binding mode. Phe-325 is on the substrate path post the active site, so the role of Phe-325 may be to help stabilize the C-terminus of the substrate.

## The LbcA supercoiled TPR domain fluctuates between a proximal and a remote state

The LbcA density in the CtpA–LbcA EM map thickens in the middle of the TPR region (Appendix Fig. S4a), suggesting that the TPR domain may fluctuate in multiple conformations. Therefore, we performed a heterogeneous refinement leading to two CtpA–LbcA EM maps. As expected, the two CtpA hexamers in the two 3D maps are superimposable but the two LbcA supercoils are wide apart, with one supercoil in a higher position remote to the front CtpA and the other in a lower position proximal to the front CtpA (Appendix Fig. S4b; Appendix Table S3). We next performed focused refinement in the top one-third region of the two maps and derived two local EM maps at around 4 Å resolution (Appendix Figs. S7 and 8; Fig. 5A–D, Appendix Table S2). We used the CtpA–LbcA atomic model described above and the TPR domain of the published LbcA crystal structure as references to build atomic models in these two new EM maps. We found that in these two local structures, the LbcA adaptor domain (H1–H6) binds to CtpA in essentially the same binding mode, and the difference between the two structures is limited to the LbcA TPR domain, which is 44 Å and 18 Å, respectively, above the front CtpA (Fig. 5B,D,E).

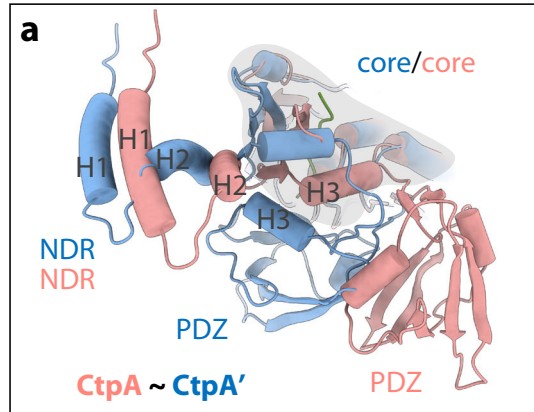

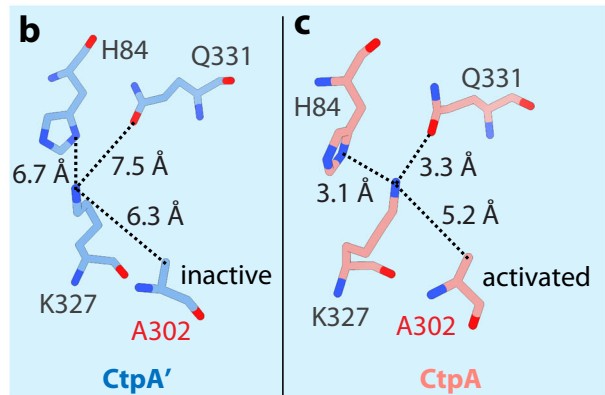

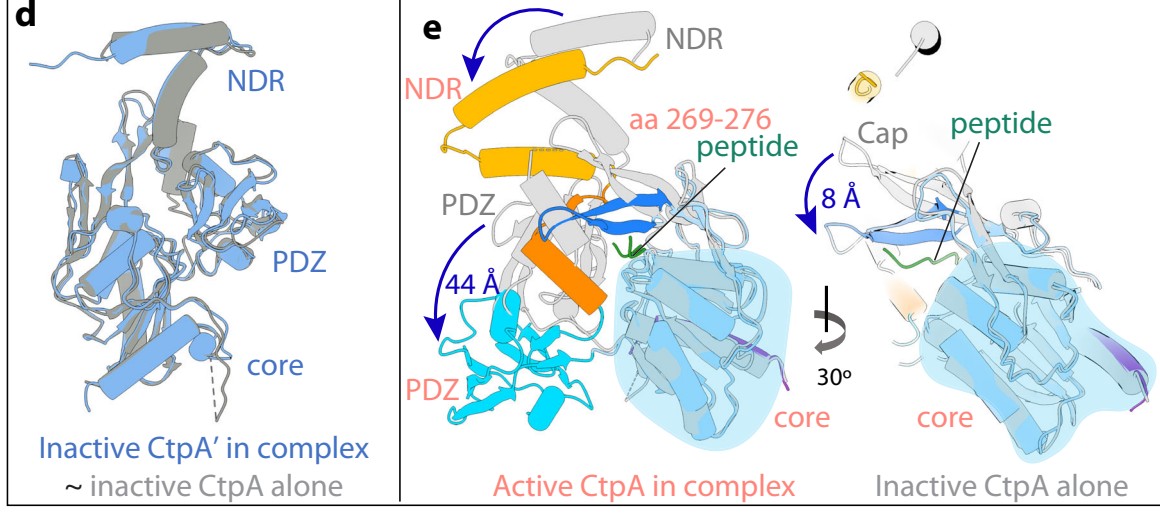

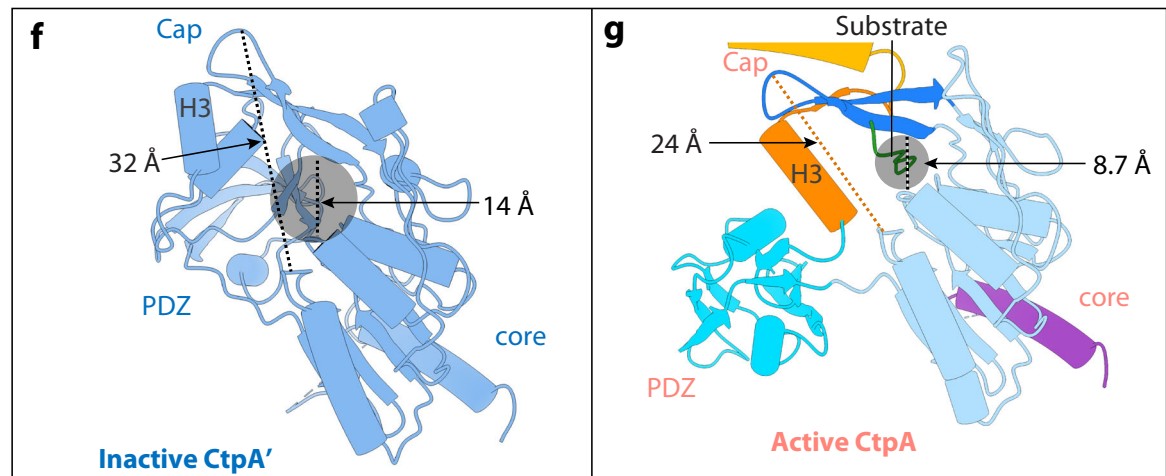

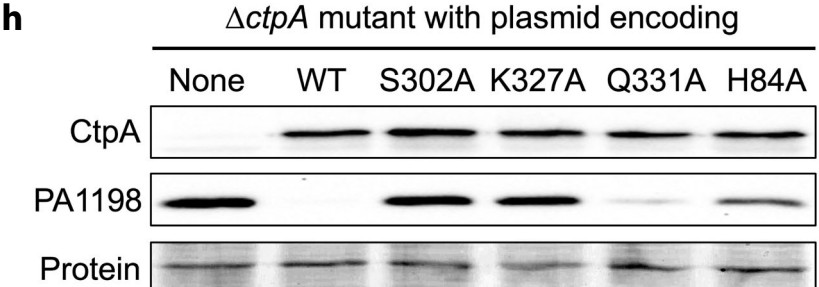

**Figure 3. Conformational changes that activate one of the two CtpA.**

(A) The front CtpA (salmon) is superimposed with the back CtpA' (light blue) by aligning their respective protease core domains, revealing drastically different structures. (B, C) Key residues in the CtpA' such as His-84, Lys-327, and Gln-331 are too far from each other for catalysis, but these residues in CtpA are arranged within catalysis distance. The catalytic residue configuration suggests that the front CtpA is active and the back CtpA' is inactive. The 5.2 Å distance is due to the S302A mutation. (D) The inactive CtpA' in the CtpA–LbcA complex resembles CtpA in the CtpA-alone hexamer. (E) Superimposition of the active CtpA with the inactive CtpA in the CtpA-alone hexamer by aligning the core domains. The NDR, PDZ, loop 269–276, and cap in the active CtpA have undergone extensive conformational changes. (F) Cartoon view of the inactive CtpA' showing an enlarged substrate binding pocket encircled by the H3, cap, core domain, and PDZ. (G) Cartoon view of the active CtpA showing the H3 shifts toward the core domain and pushes the PDZ towards the front. The cap lowers down by 8 Å to narrow down the substrate pocket and bind the substrate peptide. (H) Effect of CtpA mutations on substrate degradation in vivo. CtpA and PA1198 were detected by immunoblot with polyclonal antisera; loading was monitored by Ponceau S total protein staining of the nitrocellulose membrane used for detection (protein). Data are one representative, but at least three biological replicates of each strain have been analyzed in the laboratory. Source data are available online for this figure.

In the proximal binding mode, the LbcA hinge region (H7–H8) rotates clockwise by 26°, compared to the hinge region in the remote mode, to lower the TPR domain by 22 Å and orient it towards, but apparently not contact, the substrate binding pocket of the front active CtpA (Fig. 5F). TPRs often serve as a scaffold to mediate protein–protein interaction (Blatch and Lassle, 1999). Therefore, we suggest that the observed TPR domain movement might be linked to substrate protein binding and delivery to the CtpA proteolytic site.

## Upon LbcA binding the CtpA dimer undergoes coordinated changes to maintain the CtpA hexamer assembly

In the CtpA structure, the hexamer assembly involves the NDR–NDR interaction within a CtpA dimer and the CDR-CDR interaction between two CtpA dimers (Hsu et al, 2022). However, as we have described above, LbcA binding induces large changes in the front CtpA of the CtpA dimer (Fig. 3E). This raises the question of how the CtpA hexamer architecture is maintained upon LbcA binding. To address this question, we further refined the class IV EM map in which all three LbcA molecules are on the front side of the CtpA hexamer (Fig. 1D), which enabled the application of the C3 symmetry to improve the map quality and resolution. We improved the map to an overall resolution of 3.9 Å (Fig. 6A; Appendix Figs. S2 and S9). We found that the movements of the PDZ domain, the core domain, and the NDR of the active CtpA in the LbcA-bound CtpA hexamer are comparable to those observed in the locally refined LbcA–CtpA dimer complex (Fig. 3A,D,E). However, the threefold symmetric complex structure reveals that the CDR helix H8 (aa 419–433) of the active CtpA shifts up by one helical turn, in response to the large changes in the upper region (Fig. 6B). These changes help maintain the trimer-of-dimers configuration of the CtpA complex bound to three LbcA.

## The LbcA₃CtpA₆ configuration contributes to protease activation by LbcA

We previously showed that mutating the CDR region abolishes CtpA hexamerization and that the CtpA dimer alone had significantly reduced function (Hsu et al, 2022). This is puzzling because the CDR is far from the active site, and the CtpA dimer appears to be the functional unit. The solution of the threefold symmetric CtpA–LbcA complex now allows us to rationalize the functional importance of CtpA hexamerization. We found that the long loop (aa 376–411) that is disordered in the CtpA-alone hexamer structure (Hsu et al, 2022) becomes ordered in the

LbcA–CtpA complex structure, and we now call it the trans-interacting loop (TIL) (Fig. 6C). In the back view of the complex structure, the TIL loop from the inactive CtpA' of the lower left CtpA dimer projects upward and inserts into a space in the top CtpA dimer between the PDZ of the inactive CtpA' and the core domain of the active CtpA (Fig. 6C). Analysis of the structural features of TIL showed that the TIL Glu-385' and Asn-394' hydrogen bond with the active site His-84 and Gln-360, respectively (Fig. 6D,E). As we had already shown that His-84 facilitates CtpA activity (Fig. 3C), the mutation of its H-bond partner Glu-385' (E385A) also reduced CtpA activity both in vivo and in vitro (Fig. 6F,G). In contrast, Gln-360 is far away from the active site (Fig. 6E) and mutating the Gln-360-interacting Asn-394' to alanine (N394A) had no obvious effect on CtpA-dependent degradation in vivo (Fig. 6F). In addition to the polar interactions, the TIL residues Phe-383' and Leu-388' form a hydrophobic cluster with Phe-325 of the activated CtpA and Pro-147 of the inactive PDZ. The functional importance of the TIL Phe-383' and Leu-388' may reside in the fact that their interacting residue Phe-325 is close to the catalytic site and participates in substrate peptide binding (Fig. 4C). Interestingly, an L388A mutation reduced proteolytic activity in vivo, whereas a L388M mutation did not (Fig. 6F). It is possible that the larger methionine side chain mimics the leucine side chain better than the small alanine side chain (Fig. 6D). On the other hand, the F383A mutation had no effect, suggesting that only residues close to His-84 and Phe-325 affect the activity (Fig. 6D). Finally, deletion of the whole TIL showed similar activity to those of single and double mutations, suggesting the Glu-385 and Leu-388 may be the primary TIL residues in modulating the CtpA activity (Fig. 6G). The TIL sequence is specific to CtpA and is absent in the closest homolog CtpB. These mutational analyses indicate that the TIL loop may enhance CtpA function. However, all TIL mutations caused only a minor loss of the protease activity (Appendix Fig. S10), suggesting that the TIL loop and hexamerization of CtpA may have functions beyond regulating the protease activity.

We previously showed that CtpA alone formed hexamers in the crystal lattice (Hsu et al, 2022) and on ice at 4 °C (Appendix Fig. S12a). However, only dimeric CtpA was observed by negative staining EM where the EM grids were prepared at room temperature (Appendix Fig. S12b). Therefore, we next examined the CtpA oligomerization state by incubating the protein at 30 °C to mimic the cellular environment. Surprisingly, most CtpA became dimeric in gel filtration after 15 min of incubation at 30 °C (Fig. 7A). Further, the gel filtration peak profile of the purified co-expressed CtpA–LbcA complex also broadened upon 30 °C incubation (Appendix Fig. S12c). We next performed isothermal

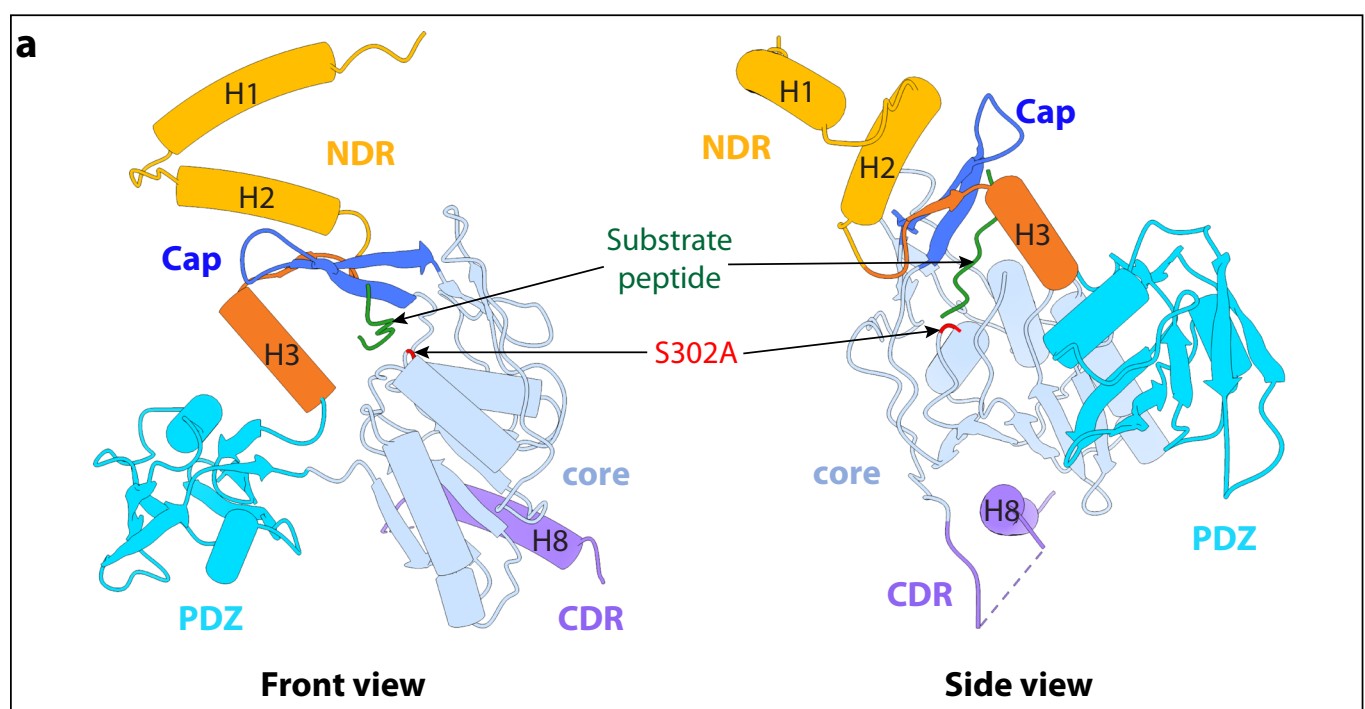

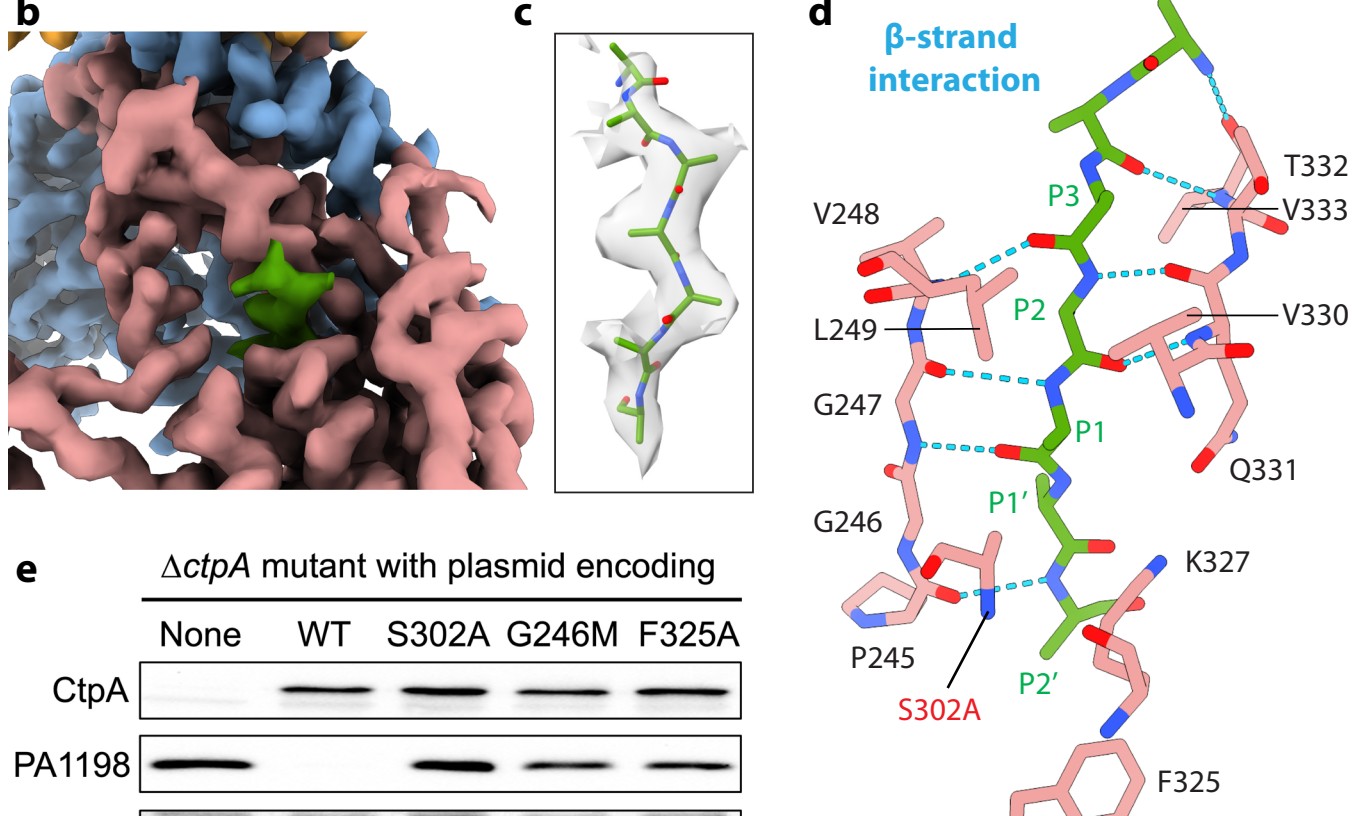

**Figure 4.  Substrate binding in the active CtpA.**

(A) The active CtpA structure in a front and side cartoon view. The bound substrate peptide chain is colored green, and the catalytic mutation S302A is colored red. (B) The EM map of the protease core region. The inactive CtpA′ is in light blue, the active CtpA is in salmon, and the co-purified substrate peptide density is in top view and colored green. (C) A side view of the substrate peptide density superimposed with the seven-alanine atomic model in sticks. (D) The substrate binding with the CtpA substrate tunnel involves multiple backbone H-bonding (dashed cyan lines), resembling that of β-strand interactions. (E) Effect of CtpA mutations on PA1198 substrate degradation in vivo. CtpA and PA1198 were detected by immunoblot with polyclonal antisera; loading was monitored by Ponceau S total protein staining of the nitrocellulose membrane used for detection (protein). Data are one representative, but at least three biological replicates of each strain have been analyzed in the laboratory. Source data are available online for this figure.

titration calorimetry (ITC) to measure the thermodynamics of LbcA binding to CtpA(S302A) at 25 °C. The titration peaks fit well with a one-site curve, i.e., one LbcA binding to one CtpA dimer (Fig. 7B), consistent with the above-described finding that CtpA is primarily a dimer at room temperature. This is further supported by the observation that a three-site fitting curve does not fit the titration peaks (Appendix Fig. S11a), as three binding sites exist only in the CtpA hexamer, which is a minor species in solution prior to LbcA binding. The binding affinity ($K_D$) between LbcA and CtpA(S302A) is $0.76 \pm 0.16\ \mu M$, similar to the affinity between the wild-type CtpA and LbcA ($K_D = 0.92 \pm 0.18\ \mu M$) (Fig. 7B; Appendix Fig. S11b), and also similar to the exclusively dimeric CtpAΔC6 ($K_D = 0.85 \pm 0.19\ \mu M$) (Appendix Fig. S11c), in further agreement with the observation the CtpA is primarily dimer at room temperature.

We further examined in vitro LbcA binding to CtpA by gel filtration by adding increasing amount of LbcA to CtpA (6:1, 6:2, and 6:3 molar ratio mixes). The addition of LbcA led to a small peak 1 representing the fully assembled CtpA hexamer bound to three LbcA complex ($LbcA_3CtpA_6$) and an increasing but gradually shifting peak 2 (Fig. 7C). The small peak 1 indicates that in vitro assembly of the $LbcA_3CtpA_6$ by mixing separately purified proteins is inefficient, and indeed, this was the reason why we resorted to co-expression to produce the $LbcA_3CtpA_6$ complex for structural analysis. SDS-PAGE analysis of these peaks revealed a constant molar ratio of 2:1 for CtpA to LbcA (Fig. 7D). Therefore, the shifting peak 2 of the three mix ratios may represent various oligomerization states of the LbcA-bound CtpA dimer, e.g., $1xLbcA_1CtpA_2$, $2xLbcA_1CtpA_2$, or $3xLbcA_1CtpA_2$, and suggests that a LbcA-bound CtpA dimer can only assemble with other LbcA-bound CtpA dimers, but not with CtpA dimer in the absence of LbcA. This scenario is consistent with the observation that the in vitro CtpA activity increases with the amount of added LbcA, rather than being saturated at 6:1 mix ratio were the CtpA hexamer assembles prior to LbcA binding (Appendix Fig. 12d). We further examined the LbcA binding with the exclusively dimeric CtpA mutant CtpAΔC6. The LbcA-bound CtpAΔC6 dimer eluted as a sharp peak, and the peak position did not shift by varying the amount of added LbcA (Appendix Fig. 13a). Indeed, 2D class averages of the cryo-EM images revealed only the CtpA dimer, individual LbcA, and the LbcA bound CtpA dimer, but neither CtpA hexamer nor the $LbcA_3CtpA_6$ complex (Appendix Fig. 13b). Therefore, LbcA binding to the CtpA dimer induces CtpA hexamerization, although in vitro assembly of the full complex of $LbcA_3CtpA_6$ is inefficient.

## Discussion

We have determined the cryo-EM structure of the CtpA–LbcA complex, revealing a protease activation mechanism very different from functionally related bacterial CTPs such as the *B. subtilis* CtpB and the *E. coli* Prc (Chueh et al, 2019; Mastny et al, 2013; Su et al, 2017). Of these two, CtpB is the closest homolog of CtpA. Their 39% amino acid sequence identity leads to comparable folds of their NDR and core domains in the monomer, but their functional oligomeric configurations are notably distinct. CtpB forms a ring-like dimer mediated by the N-terminal dimerization motif, which is similar to the NDR of CtpA, and by the C-terminal dimerization motif, in a N-to-N and a C-to-C configuration (Mastny et al, 2013). CtpB activation is controlled by the PDZ domain. In the inactive state, the PDZ domain serves as an inhibitor to block the protease tunnel. Upon substrate binding to the PDZ domain, it relocates and switches CtpB to the active form by allowing access to the proteolytic site, and by realigning the residues of a catalytic triad (Mastny et al, 2013). *E. coli* Prc assembles into a tetramer with its adapter protein NlpI in a 2:2 molar ratio (Su et al, 2017). Prc itself is a self-compartmentalized monomer with a bowl-shaped barrel and a lid-like PDZ domain. NlpI is a homodimer with an N-terminal lipidation site for anchoring to the outer membrane, and four TPR repeats for interacting with Prc and possibly with substrates. However, binding of NlpI is not required for activation of Prc. Instead, like CtpB, substrate binding to the PDZ domain of Prc triggers a structural remodeling to open the PDZ lid and to realign the residues of catalytic dyad (Chueh et al, 2019). Therefore, binding of substrate to the PDZ domain is essential to initiate the activation processes of both CtpB and Prc.

In contrast to CtpB and Prc, the activation of CtpA is initiated by LbcA binding to the NDR of CtpA. In the activated structures of CtpA, CtpB, and Prc, a co-purified peptide was found in the substrate binding tunnel (Mastny et al, 2013; Su et al, 2017). The co-purified peptide also bound to the PDZ domains of CtpB and Prc, but we could not determine if the co-purified peptide bound to the CtpA PDZ, due to the high flexibility and low resolution of this region. However, as no peptide was present in the structures of CtpA alone, or the inactive CtpA′ of the CtpA–LbcA complex, any peptide binding to the activated CtpA must take place only after LbcA binding. Therefore, binding of LbcA is the first step required to activate CtpA. In support of this, CtpA cannot form a stable binary complex with a substrate in the absence of LbcA (Chakraborty and Darwin, 2021). In contrast, LbcA can form a complex with CtpA or with a CtpA substrate independently, meaning that LbcA is likely to act as a scaffold protein to recruit CtpA and its substrates independently to form a ternary proteolytic-active complex (Chakraborty and Darwin, 2021). Indeed, LbcA has an N-terminal adaptor domain for binding to CtpA, and a separate C-terminal TPR superhelical domain that is likely to bind substrates. We propose the following activation mechanism for CtpA (Fig. 8). In the absence of LbcA, CtpA is inactive, likely in a dynamic equilibrium between a dimer and a

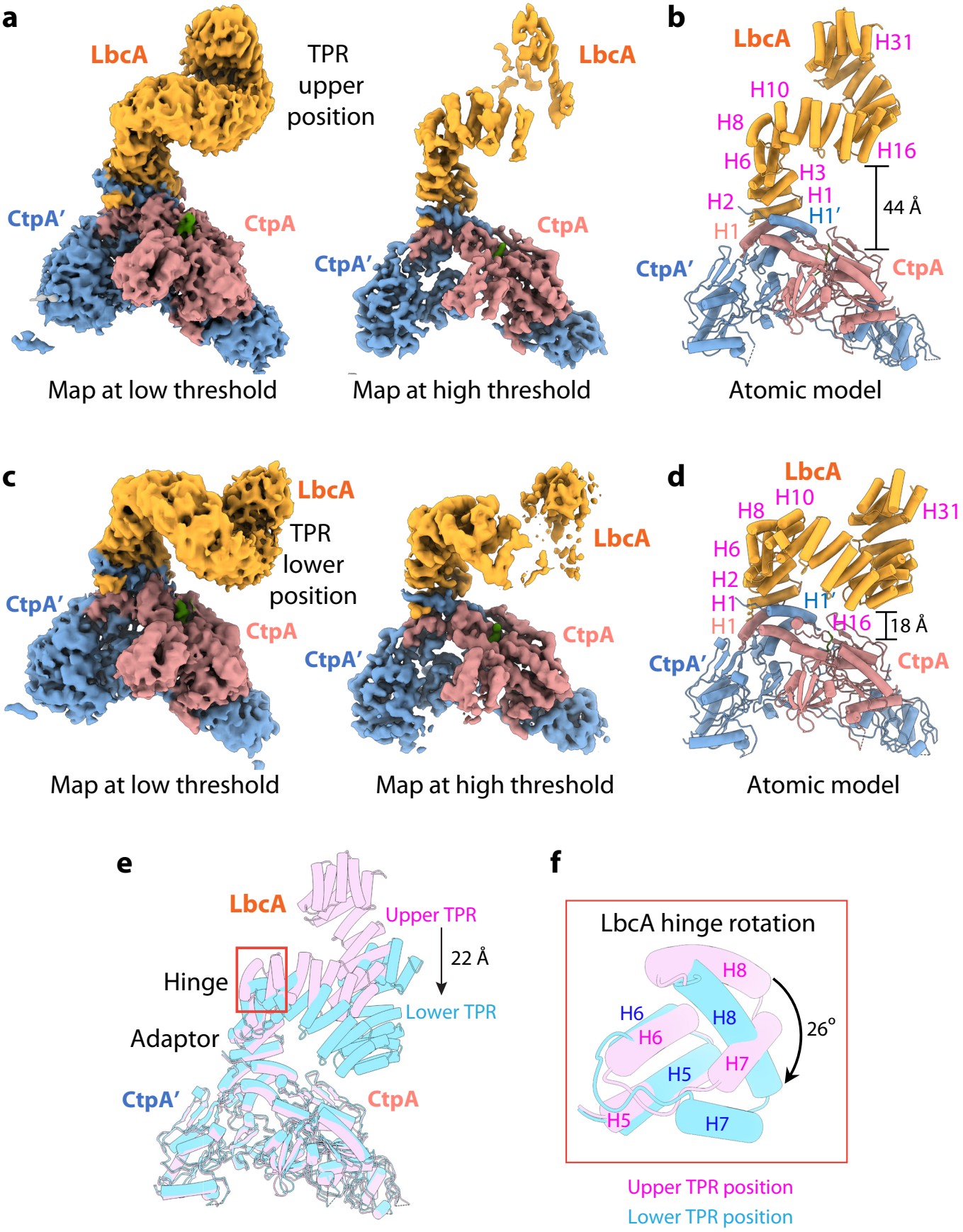

**Figure 5. The two LbcA conformations of the CtpA–LbcA complex.**

(A) Front view of the EM map of the first conformation rendered at a low (left) and high (right) threshold showing the whole LbcA TPR domain. (B) Cartoon front view of the atomic model showing the LbcA TPR is 44 Å above the CtpA in the remote binding mode. (C) Front view of the EM map of the second conformation rendered at a low (left) and high (right) threshold showing the whole LbcA TPR domain. (D) Cartoon front view of the model showing the LbcA TPR is 18 Å above the CtpA in the proximal binding mode. (E) Superimposition of the two conformations by aligning the CtpA dimer region, showing the 22 Å down-shift of the LbcA TPR in the second conformation. (F) Enlarged view of the hinge domain marked by the orange box in (E). The hinge domain rotates 26° to bring the TPR domain down by 22 Å in the second conformation.

hexamer, with dimer the predominant species. LbcA recognizes the substrates and delivers them for degradation by attaching its N-terminal adaptor domain onto the NDR of a CtpA dimer. LbcA binding triggers a series of conformational changes in CtpA, including trimerization of the LbcA-bound dimer, a relocation of the PDZ domain, and a clamping down of the cap toward the catalytic pocket. The PDZ of inactive CtpA and the core of active CtpA in the dimer further interact with the trans-interacting loop of a partner CtpA dimer. These overall conformational changes realign the catalytic residues into an active configuration and convert the wide CtpA substrate binding pocket into a narrow substrate binding tunnel. The substrate peptide is then sandwiched between two β-strands and locked in place. The realigned and now active catalytic residues then launch a nucleophilic attack at a peptide scissile bond to hydrolyze the substrate protein.

In the dimeric CtpB and tetrameric Prc–NlpI complexes, both monomers of CtpB or Prc are activated after substrate binding. In contrast, despite the interactions of LbcA H1 with both the front and the back CtpA H1 helices, only one CtpA monomer (front) is activated in each of the CtpA dimers that make up the hexamer. The superhelical TPR domain is right-handed, which means that when the LbcA adaptor domain binds to a CtpA dimer, it brings the LbcA TPR domain closer to one of the two core domains of the CtpA dimer, depending on the binding direction of the LbcA H1 helix. Also, it is always the CtpA closest to the TPR domain that gets activated, suggesting TPR domain may participate in the later proteolytic process after activation. The LbcA-bound substrate might then access the CtpA PDZ domain and prompt the downstream activation process of CtpA. Therefore, while binding of LbcA to CtpA initiates protease activation, the subsequent transfer of the substrate C-terminal region from LbcA to the PDZ domain of CtpA might still be required to complete the activation process.

It is also important to note a possible variation on the model in Fig. 8, because LbcA and CtpA might always be in complex in vivo. Indeed, CtpA fractionates with the membrane fraction in *lbcA*⁺ cells but is in the soluble periplasm in Δ*lbcA* cells, and when CtpA or LbcA are purified from *P. aeruginosa*, they bring a lot of the other one down with them (Srivastava et al, 2018). In this scenario, substrate-specificity of a constitutively active CtpA–LbcA complex might arise because only LbcA-bound substrates can be processed by CtpA. Alternatively, a final activation step of CtpA might only occur when its PDZ domain engages a substrate C-terminus presented by LbcA, as mentioned above.

There are several striking differences between the CtpA–LbcA and Prc–NlpI systems. CtpA and Prc are in divergent CTP sub-groups. LbcA is much larger than NlpI, and their primary sequences are not homologous. Even so, CtpA and Prc both use a membrane-anchored adaptor protein to bring them together with a substrate. However, CtpA is a hexamer attached to monomeric LbcA anchored on the outer membrane, whereas Prc is a monomer

attached to dimeric NlpI on the outer membrane. Therefore, the geometry of the CtpA–LbcA complex is determined by the protease CtpA, whereas that of Prc–NlpI is determined by the co-factor NlpI instead.

We also investigated the catalytic residues of CtpA in this study. In the MEROPS peptidase database, CtpA belongs to the S41A family of C-terminal processing peptidases, which are characterized by an active site containing a Ser–Lys catalytic dyad and the presence of a PDZ domain (Rawlings et al, 2002). Previously, Ser-309, Lys-334, and Gln-338 were proposed to be the residues of a catalytic triad of CtpB (Mastny et al, 2013). Based on sequence homology, Ser-302, Lys-327, and Gln-331 would be the corresponding residues in CtpA. However, although Gln-331 is conserved in the S41A family and is located close to Lys-327 in the CtpA structure after activation, a Q331A mutation had only minor impact on protease activity in vivo, showing that Gln-331 is not an essential member of a CtpA catalytic triad. Our structural analysis identified His-84 as an alternative to Gln-331 as the acid residue. Using histidine as the acid of a catalytic triad does occur in other proteases, for example, the human cytomegalovirus protease has two histidine residues, one as the base and the other as the acid, in the catalytic triad (Chen et al, 1996; Shieh et al, 1996). In other proteases, the acid is less important for activity, and they use a catalytic dyad. For example, in the structure of *E. coli* Lon Protease, a Ser–Lys dyad was identified in its active site (Botos et al, 2004). A Ser-302–Lys-327 dyad or a His-84–Ser-302–Lys-327 triad are both plausible in defining the CtpA active site. However, it should be noted that while S302A and K327A mutations eliminated CtpA activity in vivo, the H84A mutation reduced but did not eliminate activity.

We reported previously that CtpA C-terminal mutants profi-cient in dimer formation, but deficient in hexamer formation, have significantly reduced activity (Hsu et al, 2022). Here we have discovered a trans-interacting loop (TIL) that helps to explain those findings. The TIL is disordered in the CtpA-alone hexamer. It becomes ordered in the LbcA–CtpA complex and inserts into the space between the PDZ of one monomer and the core domain of the other monomer in a neighboring CtpA dimer, thereby interacting with the active configuration of that CtpA. However, removing the TIL did not reduce CtpA protease activity significantly. As the TIL is specific for CtpA and is ordered only in the CtpA hexamer bound to three LbcA, it is likely that the loop's function is primarily related to the active CtpA hexamer. The TIL concatenates all six CtpA subunits and the requirement of CtpA hexamerization remains to be investigated.

We also found two LbcA conformations with different TPR positions in the CtpA–LbcA complex. However, both conformations were in complex with active CtpA, implying that the two positions of the superhelical TPR domain have no effect on the allosteric activation mechanism. While the N-terminal adaptor is required for CtpA activation, the different TPR positions facilitate delivery of various substrates to activated CtpA. Currently, five CtpA substrates have been identified, ranging from ~17 to 50 kDa in size (Chakraborty and

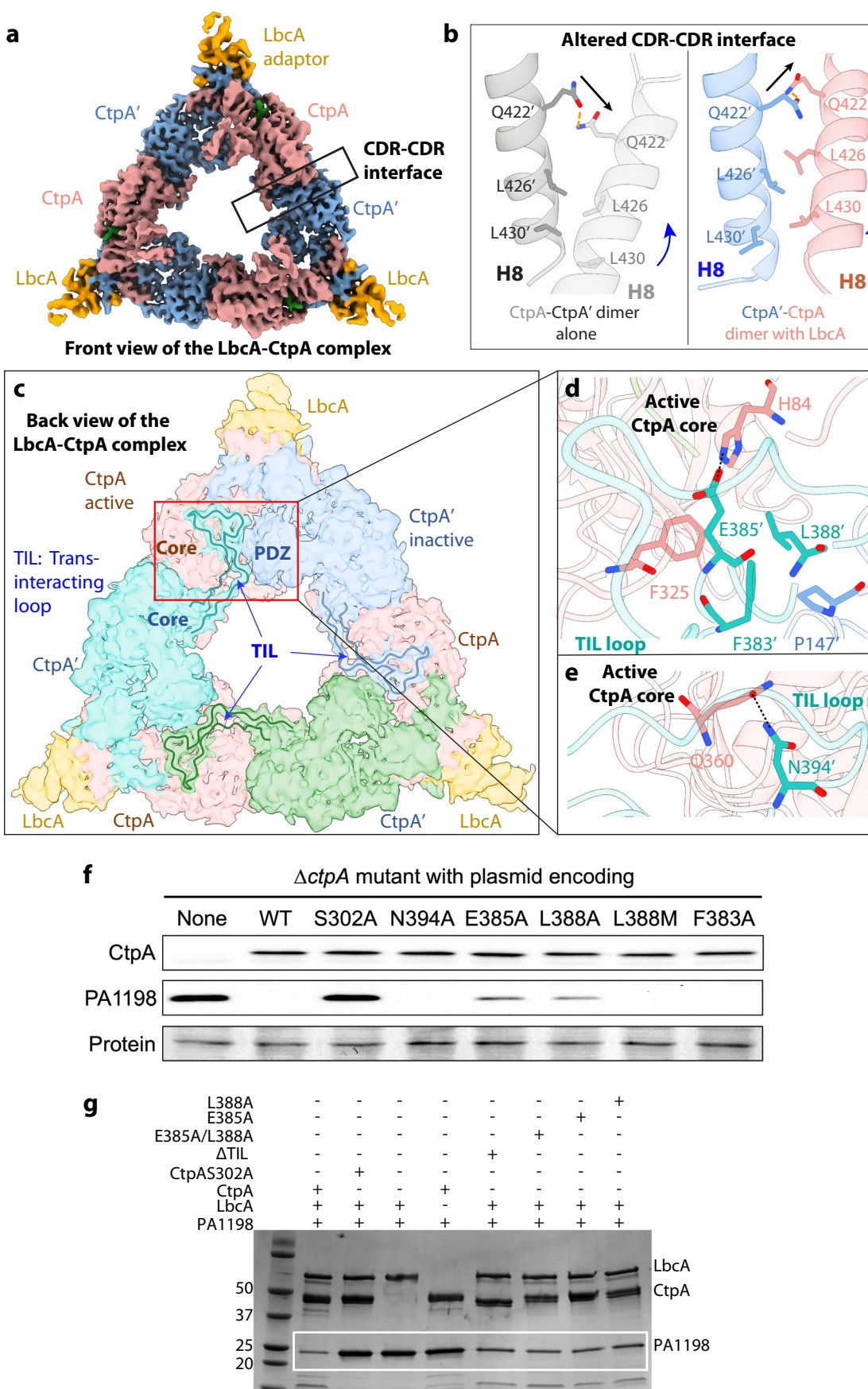

◄ **Figure 6.  Trans regulation between CtpA dimers within the CtpA hexamer.**

(A) Front view of the EM map of the threefold symmetric CtpA–LbcA complex. Only the adaptor domain of LbcA is resolved, and the more flexible TPR is averaged out. (B) A side-by-side comparison of the CDR H8 in CtpA-alone hexamer and in LbcA-bound complex, revealing an upward shift of one helical turn by the H8 of the active CtpA, altering the CDR H8–H8 interface. (C) Back view of the 3D EM map of the threefold symmetrical CtpA–LbcA complex. Superimposed on the map are the trans-interacting loops (aa 376–411; TIL) from the three back and inactive CtpA′ (cyan, green, and light blue) shown in cartoons. The red square highlights the TIL loop of the lower left CtpA dimer inserting into a gap between the top CtpA dimer. (D, E) Hydrogen bonding (dashed black lines) and hydrophobic interactions between the active CtpA core (salmon) in one CtpA dimer and the TIL loop from the inactive CtpA′ of the neighboring (lower left) CtpA dimer (cyan). (F) Effect of CtpA TIL loop mutations on substrate degradation in vivo. CtpA and PA1198 were detected by immunoblot with polyclonal antisera; loading was monitored by Ponceau S total protein staining of the nitrocellulose membrane used for detection (protein). Data are one representative, but at least three biological replicates of each strain have been analyzed in the laboratory. (G) Effect of CtpA TIL loop mutations on substrate degradation in vitro. The purified PA1198 served as the substrate for CtpA. Quantification of the relative activity is shown in Appendix Fig. S10. Source data are available online for this figure.

Darwin, 2021; Srivastava et al, 2018). To bring the C-termini of these substrates to the CtpA substrate binding tunnel, the TPR domain of LbcA needs to be configured flexibly to accommodate the substrates with different sizes. However, it is unclear how substrates engage in a CtpA–LbcA–substrate ternary complex, and the role of TPR domain in substrate binding and feeding to CtpA remains to be studied.

In summary, the structure of the CtpA–LbcA complex reveals a new activation mechanism for a bacterial CTP. In contrast to the activation mediated only by binding of a substrate to the PDZ domain of the protease, CtpA activation requires the LbcA adaptor to bind to the NDR of CtpA. In addition, trans-interaction loops between CtpA dimers within a hexamer adds a new feature to this CTP activation mechanism. Our findings are especially significant because CtpA activity is required for the normal function of the *P. aeruginosa* type III secretion system and virulence in a mouse model of pneumonia, and it also affects surface attachment, which is the first step in biofilm formation (Seo and Darwin, 2013; Srivastava et al, 2018). In addition, CtpA and some of its substrates have been linked to antibiotic resistance, including in one clinical strain, which is consistent with CtpA degrading cell wall hydrolases (Sanz-Garcia et al, 2018; Sonnabend et al, 2020). Therefore, the new insight into the CtpA activation mechanism uncovered here could help to lay the groundwork to develop new antibacterial agents to interfere with it. Finally, although we now know much more about how LbcA activates CtpA, several questions remain to be addressed regarding how this activated complex degrades its substrates. For example, how do the five known substrates of very different size and sequence bind to LbcA for engagement in the proteolytic process, and how do substrate C-termini with different sequences bind to the PDZ domain of CtpA. Perhaps different substrate C-termini interact with different residues of the PDZ domain to complete the activation process. To get more insight into these processes, the next challenge will be to obtain the structures of CtpA–LbcA–substrate ternary complexes.

# Methods

## Protein production and purification of CtpA

The construction of plasmids pET15b_CtpAΔN37 and pET15b_ CtpAΔN37S302A was described previously (Hsu et al, 2022). Briefly, the DNA sequence encoding from amino acid 38 to the C-terminus of CtpA was subcloned into a pET15b vector between the NdeI and XhoI sites. The resulting plasmid encoded the N-terminal 6xHis-tagged CtpA(ΔN37). Similar plasmids encoding CtpA(ΔN37) with S302A, ΔC6, E385A, L388A, or E385A/L388A

mutations were generated by site-directed mutagenesis using primers listed in Appendix Table S4. The TIL deletion construct (ΔTIL, F383-R412) was generated by PCR. For all CtpA proteins, *E. coli* BL21(DE3) transformants were grown at 37 °C to optical density at 600 nm $(OD_{600}) = 0.6$–0.7 before being induced for protein overproduction with 1 mM isopropyl β-D-1-thiogalactopyranoside (IPTG) at 16 °C overnight. Cells were lysed by passing through a cell disruptor in 20 mM Tris pH 8.0, 300 mM NaCl, 10 mM imidazole. The homogenate was clarified by centrifugation at 27,000×g, and the supernatant was applied to a HiTrap-Ni column (GE Healthcare) preequilibrated with the lysis buffer. Proteins were eluted with a 10–300 mM imidazole gradient in 20 column volumes (CV) of lysis buffer. Fractions containing 6xHis-CtpA were collected. All CtpA proteins were polished by gel filtration in 10 mM Tris, pH 8.0, and 150 mM NaCl using Superose 6 increase column (10 × 300 mm, GE Healthcare).

## Protein production and purification of LbcA

The DNA sequence encompassing amino acid 32 to the C-terminus of LbcA (LbcA[ΔN31]) was subcloned into the NdeI and HindIII sites of the pET24b vector. *E. coli* BL21(DE3) transformants were grown at 37 °C to $OD_{600} = 0.5$ before being induced for protein overproduction with 0.5 mM IPTG and incubated at 37 °C for another 3 h. The C-terminal 6xHis-tagged LbcA protein was purified with HiTrap-Ni in 10 mM potassium phosphate, pH 8.0, 0.25 M NaCl, and a 20-CV 10–300 mM imidazole gradient, followed by HiTrap-Q in 10 mM Tris, pH 8.0, and a 20-CV 50–500 mM NaCl gradient. Final polishing of LbcA was performed in a Superdex 200 prep-grade column (16 × 1000 mm) preequilibrated with 10 mM Tris, pH 8.0, and 150 mM NaCl.

### Protein production and purification of the CtpA–LbcA complex

Construction of plasmid pET15b_CtpAΔN37S302A was described previously (Hsu et al, 2022). A DNA fragment containing amino acids from aa 32 to the C-terminus of LbcA was subcloned into the NdeI and BglII sites of the pCDFDuet-1 vector to construct pCDF_LbcAΔN31. 6xHis-CtpA (ΔN37, S302A) from plasmid pET15b_CtpAΔN37S302A and LbcA(ΔN31) from plasmid pCDF_LbcAΔN31 were co-expressed in *E. coli* BL21(DE3). After growing the bacteria at 37 °C to $OD_{600} = 0.7$, protein production was induced by adding 1 mM IPTG and continued incubation at 16 °C overnight. The 6xHis-tagged CtpA–LbcA complex was isolated by HiTrap Ni with a 10–300 mM imidazole gradient in 10 mM potassium phosphate, pH 8.0, and 250 mM NaCl. After removing the N-terminal 6xHis-tag of CtpA using thrombin

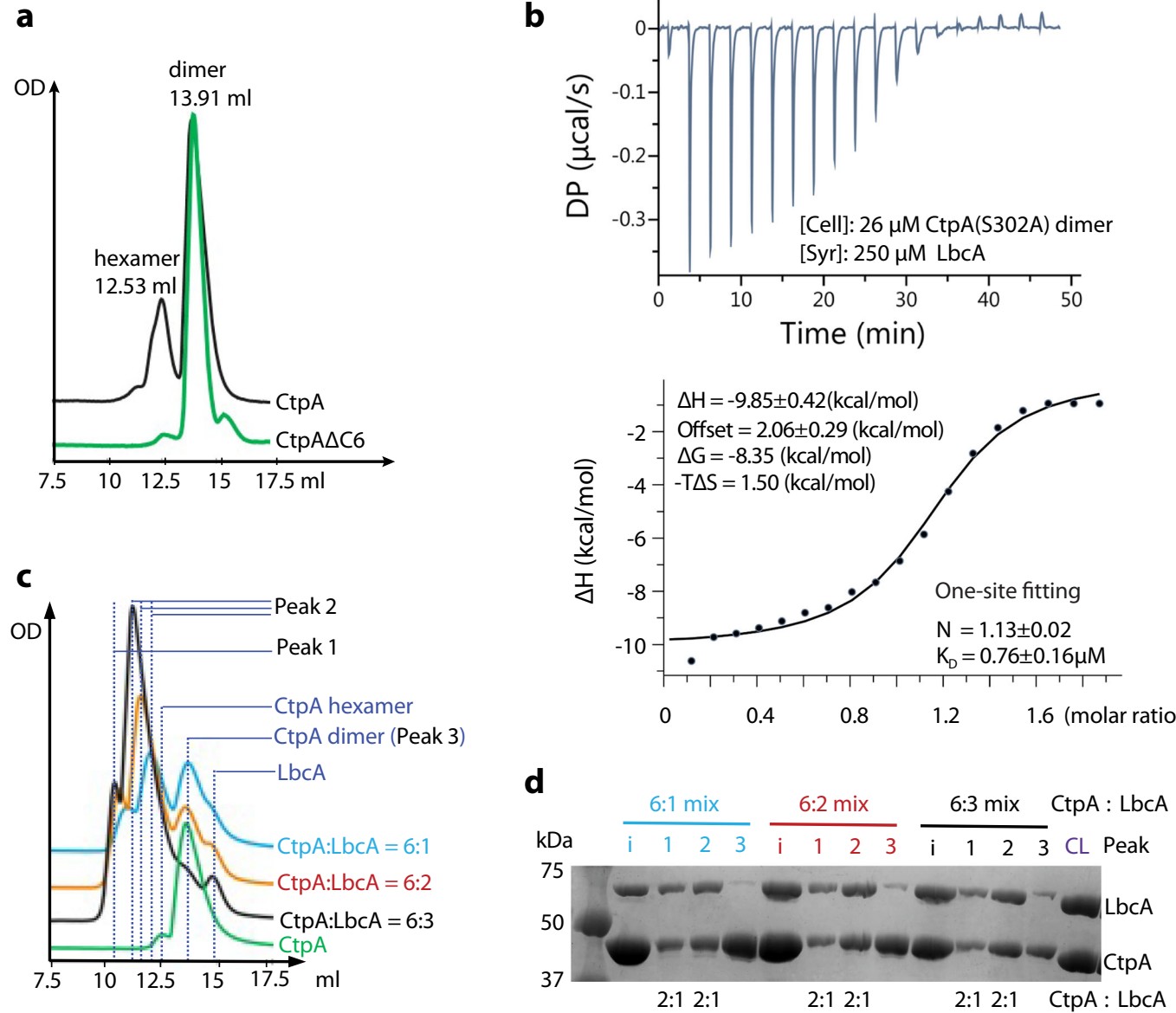

**Figure 7.  Characterization of the in vitro binding of LbcA to CtpA.**

(A) Gel filtration profiles of CtpA and CtpAΔC6 incubated at 30 °C for 15 min. CtpA is eluted primarily as dimers with hexamers as a minor species. (B) Upper panel: ITC by titrating LbcA into CtpA(S302A). Lower panel shows the fitting with the one-site binding mode, showing one LbcA bound to one CtpA dimer. (C) Gel filtration profiles of CtpA mixed at 30 °C for 15 min with an increasing amount of LbcA: with the mixing molar ratios of 6:1, 6:2, and 6:3, respectively. The relatively small peak 1 indicates the formation of a small amount of LbcA$_3$-CtpA$_6$ complex. Note the peak 2 position gradually shifts with increasing amount of LbcA. (D) SDS-PAGE gel of peaks 1, 2, and 3 from gel filtration. Peak 3 is CtpA alone, and peaks 1 and 2 have approximately the same 2:1 molar ratio (CtpA vs LbcA), resembling the 2:1 ratio of purified co-expressed CtpA–LbcA complex that is used as a control (CL). Lanes labeled with i indicate the input mixture as specified at the top (6:1, 6:2, or 6:3 mixtures). Source data are available online for this figure.

(0.5 units/mg), the CtpA–LbcA complex was further purified with a HiTrap Q in 10 mM Tris, pH 8.0, and a 20-CV 50–500 mM NaCl gradient, followed by Superose 6 Increase (10 × 300 mm) in 20 mM Tris, pH 7.5, 5 mM MgCl$_2$, 100 mM KCl, and 1 mM DTT.

## Protein production and purification of PA1198

To purify 6xHis-PA1198, *E. coli* strain M15 (pREP4) (Qiagen) containing pAJD2948 transformants were grown at 37 °C to OD$_{600}$ = 0.8 before being induced with 1 mM IPTG at 16 °C overnight. 6xHis-PA1198 was purified utilizing a HiTrap-Ni column (GE Healthcare) preequilibrated with 20 mM Tris pH 8, 200 mM NaCl. Proteins were eluted with a 10 to 300 mM imidazole gradient in lysis buffer. Fractions containing 6xHis-PA1198 were collected and further purified with HiTrap-Q in 10 mM Tris, pH 8.0, and a 50–1000 mM NaCl gradient and polished by gel filtration in 20 mM Tris, pH 8.0, and 150 mM NaCl using Superose 6 increase column (10 × 300 mm).

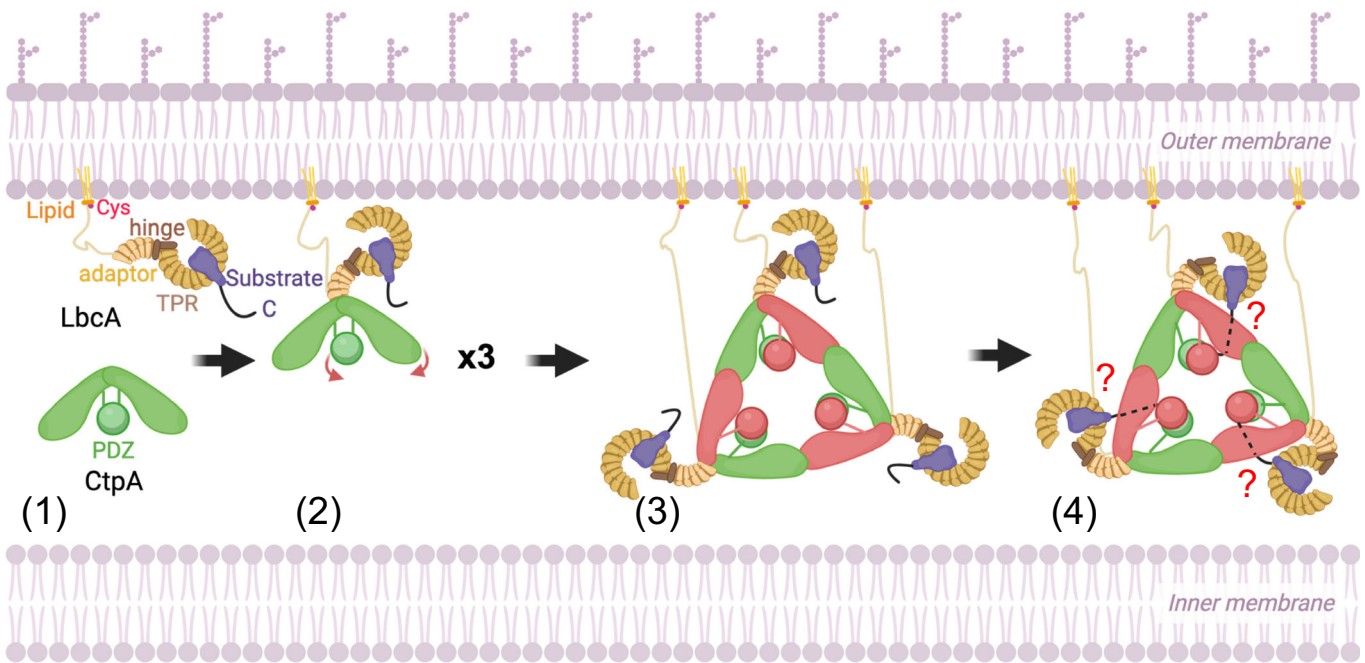

**Figure 8. A model for the LbcA-substrate dependent activation of CtpA.**

(1) The CtpA dimer is in an inactive state before LbcA binding. A substrate associates with membrane-anchored LbcA prior to LbcA binding to the CtpA dimer. C C-terminus. (2) When a substrate-bound LbcA encounters CtpA, the N-terminal adaptor domain of LbcA first attaches to the NDR of CtpA. This interaction leads to shifts of the PDZ and core domains of the CtpA that is on the same side of the bound LbcA. (3) Three LbcA-bound CtpA dimers form a triangular-shaped hexamer via their respective CDR and only one CtpA in a CtpA dimer is activated (red). (4) The substrate approaches the activated CtpA to thread the substrate C-terminus to the substrate binding tunnel by an unknown mechanism. Note the N-terminal linker of two lower LbcAs have been extended to better visualize the CtpA hexamer. The CtpA hexamer is likely held much closer to the outer membrane (three of its known substrates are membrane-attached lipoproteins).

## In vitro proteolysis assay

All in vitro proteolysis reaction mixtures contained approximately 2 μM LbcA and CtpA, and the substrate PA1198 protein was added at approximately 15 μM. Reaction mixtures were incubated at 37 °C for 3 h, and the reactions were terminated by adding SDS-PAGE sample buffer and boiling. Proteins were separated by SDS-PAGE and stained with Coomassie brilliant blue dye.

## In vitro complex formation of CtpA and LbcA

CtpA, LbcA, or mixtures of CtpA and LbcA at various molar ratios were incubated at 30 °C for 15 min in 20 mM Tris, pH 7.5, 5 mM $MgCl_2$, 100 mM KCl. After incubation, proteins were resolved in a Superdex 200 increase column (10 × 300 mm).

### Isothermal titration calorimetry (ITC)

ITC experiments were performed in a MicroCal PEAQ-ITC system (Malvern). Before performing the experiments, both CtpA and LbcA were dialyzed against 20 mM Tris, pH 7.5, 5 mM $MgCl_2$, 100 mM KCl overnight to reduce buffer effects during titration. The initial cell concentration was 9 μM of CtpA hexamer (32 μM for CtpAΔC6 dimer), and the syringe concentration was 250 μM of LbcA. The titrations were performed at 25 °C with the following parameters: 10 μCal/s reference power, 750 rpm stirring speed, 60 s initial delay, 0.4-μl first injection plus eighteen 2.0-μl sequential injections with a 6.0 s duration and a 150 s spacing.

### Negative staining EM of purified CtpA

One droplet of 3 μl CtpA(S302A) sample (10 μg/ml) was applied to a 300-mesh copper lacey carbon grid (Electron Microscopy Science) that had been freshly glow-discharged with $O_2$–air for 30 s at 30 W in a Gatan Model 950 Advanced Plasma System. After incubation at room temperature for 1 min, the protein solution was blotted with a small piece of Whatman paper. The EM grid was then washed twice with water and negatively stained twice with 1% uranium formate aqueous solution for 30 s. The grid was blotted, left to air dry, and then imaged in a 120 kV Tecnai G2 Spirit transmission electron microscope. After CTF estimation with CTFFIND4, 68,651 particles extracted from 124 raw micrographs were picked for 2D classification in Relion 3.0 [41]. The selected 2D classes of CtpA contained 23,727 particles.

### Cryo-EM sample preparation and data collection

Cryo-EM studies were conducted in the Cryo-EM facility of the Van Andel Institute. The protein concentration of the CtpA–LbcA complex used for cryo-EM study was 0.8 mg/ml. Vitrobot Mark IV (FEI, USA) was used for preparing the cryo-EM grids by setting its sample chamber environment to 6 °C and 100% humidity. One droplet of 3 μl protein sample was applied to a Quantifoil R 1.2/1.3 300-mesh Au holey carbon grid that had been freshly glow-discharged with $O_2$-air for 30 s at 30 W in a Gatan Model 950 Advanced Plasma System. After 5 s waiting time, the EM grid was blotted with a piece of 595 filter paper by setting to blotting force to 3 and blotting time to 3 s. The grid was then plunged into liquid ethane precooled with liquid nitrogen. Samples on the grid were imaged in a Titan Krios (Thermo-Fisher, USA) equipped with a

Quantum 967 energy Filter and a post GIF K3 summit direct electron detector (A.K.A. BioQuantum). Movies are collected using SerialEM software on a Gatan K3 detector operated in super-resolution counting mode with a pixel size of 0.414 Å/pixel. The movies were recorded with the objective lens focus value adjusted in the range of −1.3 to −1.8 μm and with the electron exposure dose rate set to 44 electrons/Å$^2$/s and an exposure time of 1.5 s. The total electron dose for each movie was 66 electrons/Å$^2$. A total of 26,257 movies were collected for the CtpA–LbcA complex.

### Cryo-EM data processing and 3D reconstruction

Image processing was performed in Relion 3.0 and CryoSPARC 2.8 (Appendix Fig. S2) (Punjani et al, 2017; Zivanov et al, 2018). The beam-induced motion was corrected with MotionCor2 (Zheng et al, 2017) on micrographs binned by a factor of 2 in Relion, and the motion-corrected micrographs were imported into CryoSPARC for contrast transfer function estimation using CTFFIND4 (Rohou and Grigorieff, 2015). A combination of blob picking, template picking, and Topaz were used for particle picking. The 2D averages from first 1000 blob-picked raw particle images served as the templates for template picking. A total of 2,090,314 raw particles were extracted from 25,455 micrographs and were used for reference-free 2D classification. 1,514,019 particles with good structural features were selected for ab-initio 3D reconstruction and 3D classification. To remove poorer particles, we used the C3 symmetry at this step, but applied C1 symmetry in later steps. A total of 1,185,591 particles from the two 3D classes with the correct shape and size of the CtpA hexamer were selected for 3D multi-reference heterogeneous refinement using the initial 3D model as a reference. Three classes containing a total of 971,786 particles were picked for another round of multi-reference heterogeneous refinement using an LbcA TPR removed 3D volume as the reference. Two classes with a total of 623,456 particles were combined and used for 3D homogeneous refinement followed by non-uniform refinement. The resulting final map was at 3.84 Å resolution by FSC but the better-defined region of the map was better than 3.7 Å (Appendix Fig. S3). To facilitate model building, we performed local refinement focusing on the top one-third region and improved the resolution to 3.55 Å. The same dataset particles were used to separate the maps into two conformations using the 3D volumes with different TPR positions as the references (Appendix Figs. S2 and S4). The two 3D classes were improved by 3D homogeneous refinement, non-uniform refinement, and local refinement. The final resolutions for the two maps were 3.92 Å and 4.14 Å, respectively (Appendix Figs. S7 and 8).

We noticed during image processing that the activated PDZ domain could be on either side of the CtpA hexamer, due to LbcA binding either from front or back of a CtpA hexamer. To identify all possible binding conformations, we modified the ab-initio EM map to generate references with one fixed LbcA direction at the top vertex and four different LbcA directions at the other two vertices. Heterogeneous refinement was performed using the modified reference maps, leading to four 3D classes (I to IV) (Appendix Fig. S2; Fig. 1D). In class IV, the PDZ domains were all on the same (front) side and the three LbcA all bind the CtpA hexamer in the same direction. This 3D class containing 334,527 particle images was selected for 3D homogeneous refinement and non-uniform refinement by applying the C3 symmetry. The final resolution for the threefold symmetric EM map was 3.94 Å (Appendix Fig. S9). All resolutions were estimated by applying a soft mask on the map

of CtpA–LbcA using the gold standard Fourier shell correlation (FSC) = 0.143 as the criterion (Appendix Figs. S3, S5, S7–9).

To visually confirm the dimeric assembly of CtpAΔC6 bound to one LbcA that had been revealed by the gel filtration profile (Appendix Fig. S13a), CtpAΔC6 was mixed with LbcA at 6:4 molar ratio. After incubation at 30 °C for 15 min in 20 mM HEPES, pH 7.5, 5 mM MgCl$_2$, 100 mM KCl, the samples were crosslinked with 1 mM BS(PEG)9 (PEGylated bis(sulfosuccinimidyl)suberate) on ice overnight at the concentration of 3 mg/ml, before quenching with 40 mM final concentration Quenching Buffer (1 M Tris-HCl, pH 8.0) for 15 min. The cryo-EM grid preparation procedure was the same as those used for the CtpA–LbcA complex. A small dataset of 361 movie micrographs was collected in a Talos Arctica (200 kV) equipped with Gatan K2 detector. The total electron dose for each movie micrograph was 60 electrons/Å$^2$. Image processing was performed in Relion 3.0. After motion correction with MotionCor2 and CTF estimation with CTFFIND4, 209,741 raw particle images were picked. After the first round of 2D classification, a final dataset of 60,194 particle images was retain for calculating the 2D class averages (Appendix Fig. S13b).

### Model building

The published CtpA hexamer crystal structure (PDB ID 7RQH (Hsu et al, 2022)) was used as the initial model to build the coordinates of the CtpA part of CtpA–LbcA complex, and the published LbcA crystal structure (PDB ID 7RQF (Hsu et al, 2022)) was used to build the LbcA region. We used an AlphaFold predicted CtpA PDZ domain to build the PDZ region of the EM map. All residues were manually adjusted into the EM map in Coot before the model was refined in real space in Phenix (Adams et al, 2011; Casanal et al, 2020; van Zundert et al, 2021). The model quality was estimated using MolProbity (Liebschner et al, 2019).

### In vivo CtpA activity assay

Plasmid pHERD26T derivatives encoding wild-type CtpA, CtpA-S302A, and wild-type LbcA were described previously and are listed in Appendix Table S1 (Srivastava et al, 2018). Plasmids encoding the other CtpA and LbcA mutants used in this study were made by first amplifying two separate fragments flanking the target codon by PCR. For each of these fragments, one of the primers incorporated mismatches at the target codon to convert it to encode the desired amino acid. The two fragments were joined together in a PCR splicing by overlap-extension reaction via their overlapping regions around the target codon (Heckman and Pease, 2007). These fragments were then ligated into plasmid pHERD26T using restriction sites added by the primers, as described previously (Hsu et al, 2022).

Plasmids were introduced into ΔctpA or ΔlbcA mutants by electroporation (Choi et al, 2006). Saturated cultures were diluted into 5 mL of LB broth containing 75 μg/mL tetracycline, in 18-mm diameter test tubes, at OD$_{600}$ of 0.05. Cultures were grown on a roller drum at 37 °C for 5 h. Cells harvested by centrifugation were resuspended in SDS-PAGE sample buffer at equal concentrations based on culture OD$_{600}$. Samples were separated by SDS-PAGE and transferred to nitrocellulose by semi-dry electroblotting. Chemiluminescent detection followed incubation with polyclonal antisera against CtpA, LbcA (Seo and Darwin, 2013; Srivastava et al, 2018), PA1198, or PA1048 and then goat anti-rabbit IgG horseradish peroxidase conjugate (Sigma-Aldrich catalog # A9169).

### Bacterial two-hybrid analysis

For the CtpA-T25 fusions, the wild-type *ctpA* gene without its signal sequence, or versions with mutations, were amplified by PCR and cloned into pKNT25 as XbaI-KpnI fragments. For the T18-LbcA fusions, the wild-type *lbcA* gene without its signal sequence, or versions with mutations, were amplified by PCR and cloned into pUT18C as a XbaI-KpnI fragment. Pairs of plasmids encoding these Cya-T25 and Cya-T18 derivatives were introduced simultaneously into *E. coli* BTH101 by calcium chloride transformation. Transformants were streaked onto MacConkey-maltose agar and incubated at 30 °C for ~40 h before being photographed.

## Data availability

All EM maps and associated atomic models have been deposited in the EMDB and the PDB, respectively. The accession codes are PDB 8SXE (EMD-40849) for the locally refined structure of CtpA–LbcA, PDB 8SXF (EMD-40850) for the locally refined CtpA–LbcA structure with the TPR at upper position, PDB 8SXG (EMD-40851) for the locally refined CtpA–LbcA structure with the TPR at lower position, PDB 8SXH (EMD-40852) for the threefold symmetrically refined CtpA–LbcA structure, EMD-40848 for the asymmetrically refined (C1) CtpA–LbcA EM map, EMD-40847 for the overall CtpA–LbcA EM map with the TPR at the upper position, and EMD-40846 for the overall CtpA–LbcA EM map with the TPR at the lower position.

## Peer review information

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

## Acknowledgements

The Cryo-EM dataset was collected at David Van Adel Advanced Cryo-EM suite in the Van Andel Institute. We thank Gongpu Zhao and Xing Meng for facilitating data collection. This study was supported by U.S. National Institutes of Health grant R01 AI136901 (to AD) and by Van Andel Institute (to HL). The content is solely the responsibility of the authors and does not necessarily represent the official views of the National Institutes of Health.

## Author contributions

**Hao-Chi Hsu**: Conceptualization; Data curation; Formal analysis; Validation; Investigation; Visualization; Writing—original draft; Writing—review and editing. **Michelle Wang**: Investigation; Writing—review and editing. **Amanda Kovach**: Formal analysis; Investigation; Writing—review and editing. **Andrew J Darwin**: Conceptualization; Data curation; Supervision; Funding acquisition; Project administration; Writing—review and editing. **Huilin Li**: Conceptualization; Formal analysis; Supervision; Funding acquisition; Project administration; Writing—review and editing.

## Disclosure and competing interests statement

The authors declare no competing interests.

