## [Peer Review File · The EMBO Journal]

P. aeruginosa CtpA protease adopts a novel activation mechanism to initiate the proteolytic process

Hao-Chi Hsu, Michelle Wang, Amanda Kovach, Andrew Darwin, and Huilin Li

Corresponding author(s): Huilin Li (Huilin.Li@vai.org) , Andrew Darwin (andrew.darwin@nyulangone.org)

Review Timeline:

Submission Date:	29th Jun 23
Editorial Decision:	18th Aug 23
Revision Received:	12th Dec 23
Editorial Decision:	31st Jan 24
Revision Received:	19th Feb 24
Accepted:	26th Feb 24

Editor: Hartmut Vodermaier

Transaction Report:

Prof. Huilin Li
Van Andel Institute
Structural Biology
333 Bostwick Ave NE
Grand Rapids, MI 49503

18th Aug 2023

Re: EMBOJ-2023-114888

Pseudomonas aeruginosa CtpA adopts a novel activation mechanism to initiate the proteolytic process

Dear Dr. Li,

Thank you for submitting your manuscript on *P. aeruginosa* CtpA structure and activation mechanism to The EMBO Journal. I apologize for the delay in getting back to you with a response, due to slower-than-usual referee responses at this time of the year. We have now received a complete set of reviews from three experts, which I am copying below for your information. As you will see, all referees consider your study and its findings potentially interesting, but nevertheless raise a few open questions whose answering would significantly strengthen the study. Should you be able to adequately address these queries, we would be happy to consider a revised version of the study further for publication.

Since it is our policy to consider only a single round of major revision and therefore important to comprehensively answer to all comments at the time of resubmission, I would invite you to consider the reports together with your coworkers, and to prepare a tentative response letter detailing how each of the raised criticisms/queries might be answered/clarified. On the basis of this response, we could then discuss the requirements for a successful revision already during the early stages of the revision, e.g. via email or a follow-up video call. I should add that we could also offer extension of the default three-months revision period if needed, with our 'scooping protection' (meaning that competing work appearing elsewhere in the meantime will not affect our considerations of your study) remaining of course valid also throughout this extension.

Detailed information on preparing, formatting and uploading a revised manuscript can be found below and in our Guide to Authors. Thank you again for the opportunity to consider this work for The EMBO Journal, and I look forward to hearing from you in due time.

Yours sincerely,

Hartmut Vodermaier

9) Digital image enhancement is acceptable practice, as long as it accurately represents the original data and conforms to community standards. If a figure has been subjected to significant electronic manipulation, this must be clearly noted in the figure legend and/or the 'Materials and Methods' section. The editors reserve the right to request original versions of figures and the original images that were used to assemble the figure. Finally, we generally encourage uploading of numerical as well as gel/blot image source data; for details see: embopress.org/page/journal/14602075/authorguide#sourcedata

At EMBO Press, we ask authors to provide source data for the main manuscript figures. Our source data coordinator will contact you to discuss which figure panels we would need source data for and will also provide you with helpful tips on how to upload and organize the files.

In the interest of ensuring the conceptual advance provided by the work, we recommend submitting a revision within 3 months (16th Nov 2023). Please discuss the revision progress ahead of this time with the editor if you require more time to complete the revisions. Use the link below to submit your revision:

Link Not Available

Referee #1:

In this work, Hsu et al use cryo electron microscopy to determine the structure of the CtpA protease in complex with its activator/adaptor LbcA. Prior work from the authors have shown that CtpA alone forms an inactive hexamer and the LbcA protein forms an elongated alpha-helical structure. In this current structure of the complex, they show that LbcA binds to CtpA through a rearrangement of its N-terminal adaptor domain that allows for relocation of CtpA domains that ultimately drive activation. They show that up to three LbcA molecules can bind a hexamer of CtpA, but that only half of the CtpA molecules end up being activated, while the other three CtpA monomers are needed to stabilize the active state of their partners. Interestingly, using heterogeneous refinement they show two major conformations of LbcA where the putative substrate binding domain of LbcA is up or down relative to the CtpA active site, which they suggest represents relevant states of LbcA for handing off substrates to CtpA. The paper is well written with a clear logical flow. The experimental validation of the structure is a series of western blots showing loss of a specific substrate (PA1198) when CtpA or LbcA mutants are introduced. These experiments are minimalist, but generally make the point that residues implicated by the structure to be important for the complex formation are also important for activity and are for the most part convincing. Overall, this is an excellent manuscript relating a novel structure, showing the activated state of a protease by an adaptor in an unexpected manner.

Specific comments:

1. The most interesting aspect of the LdcA/CtpA system is how LdcA facilitates degradation of select substrates by CtpA. The authors' model is that LdcA binds substrates with the TPR domains and uses the N-terminal adaptor domain to activate CtpA,

then delivers the substrates to the protease. Unfortunately, the structures do not show the TPR domains of LdcA engaged with substrates, nor do prior structures exist of LdcA bound to substrates (either via the TPR domains or elsewhere). This substrate-bound complex would address this outstanding question, but this may be beyond the scope of this work. The authors previously have shown that LdcA binds substrates, but evidence for the TPR domain being involved is speculative as far as this reviewer knows. In which case the final cartoon is a bit misleading and other models should be discussed.

2. While the structure of the complex shows how LdcA activates CtpA, it does not answer the bigger question of how LdcA delivers substrates to CtpA. If answering this question is beyond the scope of this work, the current complex opens some mechanistic questions that could be addressed without this additional information. For example, does LdcA binding to a single dimer cause activation of all other subunits (as shown in the cartoon?) or do three LdcA molecules need to bind to activate CtpA? (speaks to cooperativity of the activation) What does the presence of mixed inactive and active subunits mean for the activity as the ortholog CtpB has two active subunits in its dimeric form. Does the identity of the substrate matter (given that the only activity assay uses one substrate)? It would be ideal to have some follow-up that clarifies more about the delivery mechanism given this very nice structural work other than just the validation by mutagenesis as shown currently. That said, this structure reveals an unexpected and novel arrangement of CTP activation that is unprecedented and worthy of publication.

3. In Figure 6, the TRL loops are clearly found in the CtpA interfaces. However, the activity mutagenesis results show no (F383 and N394) or small (L388 and E385) effects from alterations on this TL loop. Additional mutagenesis (shortening or lengthening this loop) or other experiments are needed to demonstrate the importance of this region if this is to be emphasized in the manuscript.

Referee #2:

The study by Hao-Chi Hsu et al. reports cryo-EM structure of the complex of the *Pseudomonas aeruginosa* adaptor lipoprotein LbcA with the carboxyl-terminal processing protease (CTP) CtpA bound to substrate peptide. It follows on from a mBio paper by the same group describing structural analysis of CtpA hexamer alone. Major finding of this study is that binding of LbcA leads to relocation of the PDZ domain of CtpA resulting in remodeling of the substrate binding pocket. In summary, I think that this is a very exciting study, which describes a novel activation mechanism for a CTP. The manuscript is clearly and concisely written, but there are still some issues that need to be addressed before the manuscript will be acceptable for publication.

Minor concerns:

1. Fig.1b. Coomassie staining is not reliable to estimate stoichiometry, because different proteins could stain differently. Usually, Cys residue are modified with fluorescent dye and then fluorescence is quantified.

2. In the model in Figure 7, it is assumed that LbcA monomer binds to CtpA hexamer. However, the reported structure is a complex of three LbcA molecules with CtpA hexamer. Is it possible, that CtpA assembles into a tetramer with LbcA in a 6:3 molar *in vivo*? Would binding of another molecule of LbcA to the CtpA-LbcA complex affect CtpA activation? Please expand on this.

Additional suggestions:

Since CtpA controls activation of the enzymes involved in peptidoglycan remodeling, CtpA activation mechanism might be important for antibiotic resistance, virulence and biofilm formation. The reviewer feels that the authors should consider expanding discussion beyond the structural biology aspects. Discussion of significance of the discovered mechanism for development of new antibacterial agents would add more value to this study.

Referee #3:

Hsu et al. report the cryoEM analysis of the *Pseudomonas* TcpA-LbcA complex. TCPs are periplasmic protease oligomers important for the hydrolysis and therefore regulation of peptidoglycan modifying hydrolases. The OM lipoprotein LbcA plays a role in the activation of TCP and likely substrate delivery to the TCP. The structural analysis illustrates the substantial domain reorganization that results into the activation of the TCP, and reveals the presence of a bound candidate substrate in the TcpA active site. Remarkably, TcpA is an elaborate oligomeric structure, composed of a trimer of dimers, and the study reveals the presence of structural elements that stabilize the trimer, and possibly result in a cooperative stabilization of specific activation states.

The structural work is well performed and documented, and brings valuable new insights to this important field related to cell wall maintenance in Gram-negative bacteria. In its current state, the study remains a little descriptive and mechanistically

speculative. With some selected additional experiments based on the new structures obtained by Hsu et al., a more advanced functional insight on TCP regulation would be obtained, mainly in two important aspects of TcpA activity:

A) The role of the TcpA oligomeric state in its activity and regulation.

The structural analysis reveals TcpA trimer of dimers with three LbcA dimers, bound in different combinations of 'front' and 'back' conformation, and with varying map quality. The authors attribute the latter to flexibility in the different LbcA dimers. In vivo, the membrane tethering of the complex is likely to impose an all front confirmation, and may have different levels of LbcA occupancy. A number of important aspects of the oligomeric structure and its activated complex remain unresolved:

- Is it known whether activation of TcpA requires the binding of three LbcA dimers?
- Given the contact of the non-activated TcpA with a neighbouring LbcA dimers, is LcbA binding cooperative? Or is binding of a single LbcA sufficient for a cooperative activation of the three TcpA dimers (is this what is suggested in the model on Fig. 7?)?
- The authors identify a long interface loop in TcpA (i.e. aa376-424) that becomes ordered in the LbcA-bound complex and coin this as trans-regulatory loop (TRL). However, the evidence for trans regulation is only circumstantial, and does not discriminate between the structural role of the TRL in binding the neighbouring core domain (i.e. stabilizing the hexamer) or stabilizing the active conformation of the PDZ (it appears the reported structure probe core binding rather than PDZ binding).

Some relatively accessible experiments could be performed to help resolve this questions:

- o It would be interesting to include affinity measurements of the TcpA - LbcA complex formation. Ideally ITC and SPR or DPI. This would reveal if LbcA binding is cooperative.
- o Including a cryoEM collect on a 3:1 (6:2) ratio TcpA:LbcA complex could be interesting to reveal if binding of a single LbcA is sufficient to have a cooperative (partial?) activation of neighbouring TcpA dimers.
- o The authors now use an in vivo activity assay using PA1198 as readout. Could an in vitro assay be set up so that catalytic activity can be followed in function of different TcpA-LcbA stoichiometries? Using either purified PA1198, or cell lysate from the author's ctpA knockout strain. TcpA-LcbA complexes with controlled stoichiometry can then be added.

B) The role of LcbA in substrate delivery.

In their discussion and model, the authors suggest that LcbA is not only important in regulation of TcpA activity, but also in the delivery of substrates.

- In this respect, is the LcbA N-terminus (i.e. H1 - H8) sufficient to activate TcpA and to deliver substrate? Such mutant would allow the authors to test the substrate delivery hypothesis they propose in the discussion.

Minor points:

- When citing PDB access codes to structures used in the study, include the primary references, for example Line 125-126.
- Lns 131-135 (Fig. 2a). The renumbering of the N-terminal helices in LbcA alone versus LbcA-TcpA complex is confusing. Particularly since the renumbering is not done consistently, in Fig. 5f for example, it seems that the renumbering of helices is not done. Use H1 and H1' and H1" in LbcA alone versus LbcA-TcpA complex, or something equivalent (H1A / H1B etc.) would resolve this.
- Use of the same colour scheme as in Figure 4 would make Figs. 3d-g easier to follow.
- For ease of reading, it would be advisable to introduce the presence of the candidate substrate peptide earlier in the text. Fig. now shows the presence of a substrate peptide, but this is not mentioned in the legend and only introduced later in the text. That makes it initially unclear on what basis the substrate binding pocket is defined while reading the text.
- Has an attempt been made to identify the candidate substrate peptide(s) by mass spectrometry?

General introductory comments:

We are grateful to the referees for their time, and their helpful critiques. Of course, we were happy to see that our manuscript was received positively, and that the novelty of the CtpA-LbcA complex, and how LbcA activates CtpA, was appreciated.

However, the reviews did make us realize that we made the mistake of including too much commentary on how LbcA might engage with substrates in the CtpA complex, which is not a focus of this study. Although we do next plan to move on to study substrate-containing complexes, we are not yet able to do that. Our intention in this work was to report the characterization of the CtpA-LbcA complex, focusing on how LbcA interfaces with CtpA, and more excitingly, what happens as a result of that interaction to convert CtpA into an active protease. What we discovered in that respect alone is completely novel and, we think, very exciting. Therefore, in the revision we have modified the text to correct this mistake and reduced our speculation about the separate role of LbcA in engaging with CtpA substrates, and confined most of that to the Discussion section where some limited speculation is hopefully acceptable.

Referee #1:

In this work, Hsu et al use cryo electron microscopy to determine the structure of the CtpA protease in complex with its activator/adaptor LbcA. Prior work from the authors have shown that CtpA alone forms an inactive hexamer and the LbcA protein forms an elongated alpha-helical structure. In this current structure of the complex, they show that LbcA binds to CtpA through a rearrangement of its N-terminal adaptor domain that allows for relocation of CtpA domains that ultimately drive activation. They show that up to three LbcA molecules can bind a hexamer of CtpA, but that only half of the CtpA molecules end up being activated, while the other three CtpA monomers are needed to stabilize the active state of their partners. Interestingly, using heterogeneous refinement they show two major conformations of LbcA where the putative substrate binding domain of LbcA is up or down relative to the CtpA active site, which they suggest represents relevant states of LbcA for handing off substrates to CtpA. The paper is well written with a clear logical flow. The experimental validation of the structure is a series of western blots showing loss of a specific substrate (PA1198) when CtpA or LbcA mutants are introduced. These experiments are minimalist, but generally make the point that residues implicated by the structure to be important for the complex formation are also important for activity and are for the most part convincing. Overall, this is an excellent manuscript relating a novel structure, showing the activated state of a protease by an adaptor in an unexpected manner.

Specific comments:

Note that in their comments below this referee accidentally referred to LbcA as LdcA.

1. The most interesting aspect of the LdcA/CtpA system is how LdcA facilitates degradation of select substrates by CtpA. The authors' model is that LdcA binds substrates with the TPR domains and uses the N-terminal adaptor domain to activate CtpA, then delivers the substrates to the protease. Unfortunately, the structures do not show the TPR domains of LdcA engaged with substrates, nor do prior structures exist of LdcA bound to substrates (either via the TPR domains or elsewhere). This substrate-bound complex would address this outstanding question, but this may be beyond the scope of this work. The authors previously have shown that LdcA binds substrates, but evidence for the TPR domain being involved is speculative as far as this reviewer knows. In which case the final cartoon is a bit misleading and other models should be discussed.

We have shown in our published work that LbcA binds substrates. And our structures now show that the N-terminal domain of LbcA is responsible for the binding and activation of CtpA. In an attempt to address the reviewer's concern, we have now constructed a TPR truncated LbcA (LbcA-NTD with N-terminal MBP fusion). The truncation pulled down CtpA efficiently but failed to pull down the substrate PA1198 in comparison with a MBP only negative control (review Fig. 1a). The CtpA-LbcA(MBP-NTD) complex showed similar elution profile to that of CtpA-LbcA complex, indicating that

the adaptor domain alone was capable of binding and inducing the formation of trimer-of-dimer conformation of CtpA (review Fig. 1b). Moreover, the CtpA-LbcA (MBP-NTD) complex appeared to degrade excess MBP-fused LbcA-NTD in solution (Fig. 1c, red arrow) but had little protease activity towards PA1198 in comparison with the full length LbcA *in vitro*. These results indicate to us that the LbcA adaptor domain functions independently from the TPR domain and that an authentic CtpA substrate requires the assistance from the LbcA TPR domain to be degraded by CtpA (review Fig. 1c).

However, we realize that these data alone are insufficient to establish TPR's role in substrate recruitment, because none is direct evidence. Therefore, we suspect that presenting this may cause more questions/concerns than they address, and decided not to present them in the revised manuscript. But we are open to show them in the manuscript if the reviewer is in favor of this. Finally, we emphasize that obtaining the structure of substrate-bound complexes (the ternary complex of CtpA-LbcA-substrate protein) is something we hope to achieve in the future, which will ultimately resolve the issue of substrate recruitment. As the referee suggested, this is beyond the scope of this work, because this manuscript is focused only on understanding how LbcA activates CtpA.

It is known that TPR motifs are involved in protein-protein interactions. Therefore, the LbcA C-terminal TPR domain is likely responsible for the observed substrate binding activity. Based on these reasons, we have retained the substrate on the LbcA TPR in our revised model sketch (revised **Fig. 8**). We now clearly state in the figure legend that this aspect of the model is speculative.

2. While the structure of the complex shows how LdcA activates CtpA, it does not answer the bigger question of how LdcA delivers substrates to CtpA. If answering this question is beyond the scope of this work, the current complex opens some mechanistic questions that could be addressed without this additional information. For example, does LdcA binding to a single dimer cause activation of all other subunits (as shown in the cartoon?) or do three LdcA molecules need to bind to activate CtpA? (speaks to cooperativity of the activation) What does the presence of mixed inactive and active subunits mean for the activity as the ortholog CtpB has two active subunits in its dimeric form. Does the identity of the substrate matter (given that the only activity assay uses one substrate)? It would be ideal to have some follow-up that clarifies more about the delivery mechanism given this very

nice structural work other than just the validation by mutagenesis as shown currently. That said, this structure reveals an unexpected and novel arrangement of CTP activation that is unprecedented and worthy of publication.

We have performed additional gel filtration and ITC experiments to address the reviewer's comment. We now show that most CtpA exists predominately as a dimer at 30°C as compared to the primarily hexamer at 4°C (**Supplementary Fig. 10a-c**). LbcA binding to the CtpA dimer promotes trimerization of the CtpA dimer and stabilizes the CtpA hexamer. Therefore, the CtpA hexamer can only bind to 3 LbcA. However, we cannot exclude the possibility of CtpA hexamer with one or two LbcA in some special conditions as LbcA-free CtpA can still form hexamer at 4°C.

Our structural studies clearly show that only one of the two CtpA in the CtpA dimer of the CtpA hexamer is activated by LbcA, and this is different from the activation mechanisms of CtpB and Prc. Currently, we do not have a hypothesis as to why only one of two CtpA is activated.

The identity of the substrate does not matter. We show the level of the PA1198 substrate as an indicator of CtpA activity simply because it is a substrate for which we have an effective antiserum for detection of the endogenous level. When we first started generating CtpA and LbcA mutants we found that mutations that reduced degradation of one substrate also reduced degradation of another for which we also have an adequate antiserum (PA1048). No new information was gained from looking at both substrates, and so we focused on using PA1198 as the model substrate. All known CtpA substrates require LbcA for their degradation (published), and so any reduction in the ability of LbcA to activate CtpA, which is the focus of this manuscript, will interfere with all.

Delving into the substrate delivery mechanism effectively is going to need structural information of the substrate-bound complex, and as mentioned above, this is beyond the scope of this manuscript that is focused on understanding how CtpA activates LbcA. Even without going onto that next stage of the work yet, we are happy that the referee noted that "this structure reveals an unexpected and novel arrangement of CTP activation that is unprecedented and worthy of publication"

3. In Figure 6, the TRL loops are clearly found in the CtpA interfaces. However, the activity mutagenesis results show no (F383 and N394) or small (L388 and E385) effects from alterations on this TL loop. Additional mutagenesis (shortening or lengthening this loop) or other experiments are needed to demonstrate the importance of this region if this is to be emphasized in the manuscript.

To address the reviewer's comment, we have constructed several point mutations (E385A, L388A, E385A/L388A) and a TIL (TRL previously) deletion mutant (DTIL) and examined their protease activity *in vitro*. All these mutant CtpA showed comparable reduced activities. However, because the TIL mutations had only minor effects in our *in vivo* and *in vitro* protease activity assays, we have downplayed the importance of the TIL, focusing on its role in stabilizing the CtpA hexamer, rather than in activating the catalytic pocket.

Referee #2:

The study by Hao-Chi Hsu et al. reports cryo-EM structure of the complex of the *Pseudomonas aeruginosa* adaptor lipoprotein LbcA with the carboxyl-terminal processing protease (CTP) CtpA bound to substrate peptide. It follows on from a mBio paper by the same group describing structural analysis of CtpA hexamer alone. Major finding of this study is that binding of LbcA leads to relocation of the PDZ domain of CtpA resulting in remodeling of the substrate binding pocket. In summary, I think that this is a very exciting study, which describes a novel activation mechanism for

a CTP. The manuscript is clearly and concisely written, but there are still some issues that need to be addressed before the manuscript will be acceptable for publication.

Minor concerns:

1. Fig.1b. Coomassie staining is not reliable to estimate stoichiometry, because different proteins could stain differently. Usually, Cys residue are modified with fluorescent dye and then fluorescence is quantified.

We agree. We have added the SYPRO Ruby staining gel image in revision. But we are not just relying on staining; cryo-EM is arguably the best approach to establish the stoichiometry. We are simply stating that the band density is roughly consistent with the EM-based determination.

2. In the model in Figure 7, it is assumed that LbcA monomer binds to CtpA hexamer. However, the reported structure is a complex of three LbcA molecules with CtpA hexamer. Is it possible, that CtpA assembles into a tetramer with LbcA in a 6:3 molar in vivo? Would binding of another molecule of LbcA to the CtpA-LbcA complex affect CtpA activation? Please expand on this.

We did not observe a CtpA tetramer in any of our preparations. We found during revision experiments that CtpA exists as dimer and hexamer in equilibrium, and the oligomeric state is sensitive to temperature. CtpA is mostly hexameric at 4°C and predominantly dimeric at 30°C. At room temperature or more physiologically relevant 30°C, LbcA binding to a CtpA dimer induces CtpA hexamerization. Consequently, only the configuration of one hexamer with 3 LbcA exists in the complex. This is consistent with the observation that only CtpA hexamer was found in our cryo-EM images of the co-expressed CtpA-LbcA complex.

Additional suggestions:

Since CtpA controls activation of the enzymes involved in peptidoglycan remodeling, CtpA activation mechanism might be important for antibiotic resistance, virulence and biofilm formation. The reviewer feels that the authors should consider expanding discussion beyond the structural biology aspects. Discussion of significance of the discovered mechanism for development of new antibacterial agents would add more value to this study.

This is a good point. We didn't want to stray too far from discussing the structural aspects, because that is clearly the focus of this paper and the nature of the majority of the data. However, we have modified the final paragraph of the Discussion to include some consideration of the implications of our findings in terms of CtpA-dependent phenotypes.

Referee #3:

Note that in their comments below this referee accidentally referred to the protease as TcpA, whereas the name is CtpA.

Hsu et al. report the cryoEM analysis of the Pseudomonas TcpA-LbcA complex. TCPs are periplasmic protease oligomers important for the hydrolysis and therefore regulation of peptidoglycan modifying hydrolases. The OM lipoprotein LbcA plays a role in the activation of TCP and likely substrate delivery to the TCP. The structural analysis illustrates the substantial domain reorganization that results into the activation of the TCP, and reveals the presence of a bound candidate substrate in the CtpA active site. Remarkably, CtpA is an elaborate oligomeric structure, composed of a trimer of

dimers, and the study reveals the presence of structural elements that stabilize the trimer, and possibly result in a cooperative stabilization of specific activation states.

The structural work is well performed and documented, and brings valuable new insights to this important field related to cell wall maintenance in Gram-negative bacteria. In its current state, the study remains a little descriptive and mechanistically speculative. With some selected additional experiments based on the new structures obtained by Hsu et al., a more advanced functional insight on TCP regulation would be obtained, mainly in two important aspects of TcpA activity:

A) The role of the TcpA oligomeric state in its activity and regulation.

Note that in their comments below this referee refers to LbcA dimers, whereas LbcA acts as a monomer.

The structural analysis reveals TcpA trimer of dimers with three LbcA dimers, bound in different combinations of 'front' and 'back' conformation, and with varying map quality. The authors attribute the latter to flexibility in the different LbcA dimers. In vivo, the membrane tethering of the complex is likely to impose an all front confirmation, and may have different levels of LbcA occupancy. A number of important aspects of the oligomeric structure and its activated complex remain unresolved:

- Is it known whether activation of TcpA requires the binding of three LbcA dimers?

We showed in our previous publication that a mutant CtpA that only forms dimers (CtpA Δ C6) binds one LbcA and has reduced but detectable protease activity. Therefore, one LbcA is able to activate a CtpA dimer. However, we now report that in the presence of the activating LbcA, the wild type CtpA dimers are quickly converted into hexamers.

- Given the contact of the non-activated TcpA with a neighbouring LbcA dimers, is LbcA binding cooperative? Or is binding of a single LbcA sufficient for a cooperative activation of the three TcpA dimers (is this what is suggested in the model on Fig. 7)?

Thanks for the good question. We found during revision experiments that CtpA exists as dimers and hexamers in equilibrium, and the oligomeric state is sensitive to temperature. CtpA is mostly hexameric at 4°C and predominantly dimeric at 30°C. At room temperature or more physiologically relevant 30°C, LbcA binding to CtpA dimer induces CtpA hexamerization. Therefore, each CtpA dimer of the hexamer has one LbcA. We have revised the model (revised **Fig. 8**), proposing that an inactive CtpA is mostly a dimer in physiological environment. LbcA binding leads to both activation of the CtpA dimer and rapidly conversion of the dimer into the hexamer. Because LbcA binding is prior to the CtpA hexamerization, the binding of individual LbcAs to the three vertices of CtpA hexamer is non-cooperative (revised **Supplemental Figs. 9-11**, new **Fig. 7**).

- The authors identify a long interface loop in TcpA (i.e. aa376-424) that becomes ordered in the LbcA-bound complex and coin this as trans-regulatory loop (TRL). However, the evidence for trans regulation is only circumstantial, and does not discriminate between the structural role of the TRL in binding the neighbouring core domain (i.e. stabilizing the hexamer) or stabilizing the active conformation of the PDZ (it appears the reported structure probe core binding rather than PDZ binding).

Thanks for the insightful comment. To address the reviewer's concern, we have renamed the trans-regulating loop (TRL) as trans-interacting loop (TIL) to downplay the functional importance of the loop. We have produced a TIL (TRL previously) deletion mutant CtpA and found that the protease

remains partially active (revised **Fig. 6g**). The TIL is disordered in our published crystal structures of the CtpA hexamer. The TIL becomes ordered and is inserted between an inactive PDZ and an active core domain of a partner CtpA dimer in the current cryo-EM map of the CtpA-LbcA complex. Therefore, the TIL does make contact with the active core domain (**Fig. 6d-e**), but our *in vivo* and *in vitro* protease assays of the TIL deletion or mutations showed only minor effects. The role of TIL is speculative and needs further research. Therefore, we agree with the reviewer that the TIL may function primarily to stabilize the hexamer, and its role in regulating the protease activity is minor and secondary. We have revised the manuscript accordingly.

Some relatively accessible experiments could be performed to help resolve this questions:

o It would be interesting to include affinity measurements of the CtpA - LbcA complex formation. Ideally ITC and SPR or DPI. This would reveal if LbcA binding is cooperative.

We have performed the requested ITC experiment and showed that the affinity is around 1 μ M in revision (revised **Fig. 7a, Supplemental Fig. 9**).

o Including a cryoEM collect on a 3:1 (6:2) ratio CtpA:LbcA complex could be interesting to reveal if binding of a single LbcA is sufficient to have a cooperative (partial?) activation of neighbouring CtpA dimers.

Our *in vitro* assembly did not result in stable CtpA-LbcA complexes. We therefore resorted to co-expression of CtpA and LbcA in *E. coli*. The co-expressed CtpA-LbcA complex is stable and purified as CtpA hexamer with 3 LbcA. To address the reviewer's question, we have performed a gel filtration experiment of CtpA and LbcA binding at various molar ratios (6:1, 6:2, and 6:3) (new **Fig. 7c**). We identified a mixture of CtpA hexamer and dimer, and the CtpA dimer conversion to the hexamer is gradual and LbcA concentration dependent. However, we are unable to obtain homogeneous 3:1 complex for a cryoEM analysis. Furthermore, we found the CtpA protease activity is dependent on the amount of added LbcA (new **Supplemental Fig. 10d**). Because LbcA binding is prior to CtpA hexamerization in our new model, we now believe that the LbcA binding is non-cooperative, and a single LbcA can activate one of the two CtpA monomers at the vertex where the LbcA binds. We have revised the main text accordingly.

o The authors now use an *in vivo* activity assay using PA1198 as readout. Could an *in vitro* assay be set up so that catalytic activity can be followed in function of different CtpA-LbcA stoichiometries? Using either purified PA1198, or cell lysate from the author's ctpA knockout strain. CtpA-LbcA complexes with controlled stoichiometry can then be added.

Thanks for the good suggestion. We have now performed *in vitro* protease assays (**Fig. 6g, Supplemental Fig. 10d**). We have demonstrated the LbcA binding activates a CtpA dimer and converts the CtpA dimer to CtpA hexamer at various molar ratios of CtpA: LbcA (6:1, 6:2, 6:3) (**Fig. 7c**). As mentioned above, we now believe the three LbcA binding sites on CtpA hexamer are non-cooperative, as reflected by gel filtration profiles and the gradually increasing CtpA protease activity at an increasing amount of added LbcA (**Supplemental Fig. 10d**).

B) The role of LcbA in substrate delivery.

In their discussion and model, the authors suggest that LcbA is not only important in regulation of CtpA activity, but also in the delivery of substrates.

True, but more than a suggestion. We showed in a previous publication that LbcA forms separate, independent, direct interactions with the CtpA substrates, as well as with CtpA. However, CtpA cannot interact with any of its substrates in the absence of LbcA. In other words, one role of LbcA is to act as a scaffold protein for CtpA and its substrates, and so the role of LbcA in bringing the substrates close to CtpA is established.

What we did suggest/speculate now is that it is the TPR domains of LbcA that bind to substrates, given that the role of TPR domains is to mediate protein-protein interactions in other systems. This role is partially substantiated in our experiments (see below) by demonstrating that the TPR deletion LbcA can activate CtpA for degradation of an artificial substrate (MBP-fused LbcA NTD), but not for degradation of the true substrate PA1198 *in vitro*.

- In this respect, is the LbcA N-terminus (i.e. H1 - H8) sufficient to activate CtpA and to deliver substrate? Such mutant would allow the authors to test the substrate delivery hypothesis they propose in the discussion.

We have shown in our published work that LbcA binds substrates. And our structures now show that the N-terminal domain of LbcA is responsible for the binding and activation of CtpA. In an attempt to address the reviewer's concern, we have now constructed a TPR truncated LbcA (LbcA-NTD with N-terminal MBP fusion). The truncation pulled down CtpA efficiently but failed to pull down the substrate PA1198 in comparison with a MBP only negative control (review Fig. 1a). The CtpA-LbcA(MBP-NTD) complex showed similar elution profile to that of CtpA-LbcA complex, indicating that

the adaptor domain alone was capable of binding and inducing the formation of trimer-of-dimer conformation of CtpA (review Fig. 1b). Moreover, the CtpA-LbcA (MBP-NTD) complex appeared to degrade excess MBP-fused LbcA-NTD in solution (Fig. 1c, red arrow) but had little protease activity towards PA1198 in comparison with the full length LbcA *in vitro*. These results indicate to us that the LbcA adaptor domain functions independently from the TPR domain and that an authentic CtpA substrate requires the assistance from the LbcA TPR domain to be degraded by CtpA (review Fig. 1c).

However, we realize that these data alone are insufficient to establish TPR's role in substrate recruitment, because none is direct evidence. Therefore, we suspect that presenting this may cause more questions/concerns than they address, and decided not to present them in the revised manuscript. But we are open to show them in the manuscript if the reviewer is in favor of this. Finally, we emphasize that obtaining the structure of substrate-bound complexes (the ternary complex of CtpA-LbcA-substrate protein) is something we hope to achieve in the future, which will ultimately resolve the issue of substrate recruitment. As the referee suggested, this is beyond the scope of this work, because this manuscript is focused only on understanding how LbcA activates CtpA.

It is known that TPR motifs are involved in protein-protein interactions. Therefore, the LbcA C-terminal TPR domain is likely responsible for the observed substrate binding activity. Based on these reasons, we have retained the substrate on the LbcA TPR in our revised model sketch (revised Fig. 8). We now clearly state in the figure legend that this aspect of the model is speculative.

Minor points:

- When citing PDB access codes to structures used in the study, include the primary references, for example Line 125-126.

We have added the original references in revision.

- Lns 131-135 (Fig. 2a). The renumbering of the N-terminal helices in LbcA alone versus LbcA-TcpA complex is confusing. Particularly since the renumbering is not done consistently, in Fig. 5f for example, it seems that the renumbering of helices is not done. Use H1 and H1' and H1'' in LbcA alone versus LbcA-TcpA complex, or something equivalent (H1A / H1B etc.) would resolve this.

Thanks for the good suggestion. We have added a letter "c" to the labeling of helices in the crystal structure (e.g., H1c), to distinguish the helix (e.g., H1) in the cryo-EM structure. Fig. 2a-c have been revised.

- Use of the same colour scheme as in Figure 4 would make Figs. 3d-g easier to follow.

We have revised **Fig. 3e, g** as advised.

- For ease of reading, it would be advisable to introduce the presence of the candidate substrate peptide earlier in the text. Fig. now shows the presence of a substrate peptide, but this is not mentioned in the legend and only introduced later in the text. That makes it initially unclear on what basis the substrate binding pocket is defined while reading the text.

Thanks for the good suggestion. We now point out the presence of the peptide in the first presentation of the structure (revised **Fig. 1f**).

- Has an attempt been made to identify the candidate substrate peptide(s) by mass spectrometry?

Thanks for the good question. But we have not attempted to identify the peptide identity, for two reasons. 1) The proteins were expressed in *E. coli*, thus the peptide(s) found in the binding site won't be the natural substrate(s) of *P. aeruginosa*. 2) The EM map quality is good at the catalytic core domain, but no amino acid side chain densities are present, such that only a polyaniline model can be built. This is a clear indicative of random or mixed peptide sequences that have been averaged out to alanine.

Prof. Huilin Li
Van Andel Institute
Structural Biology
333 Bostwick Ave NE
Grand Rapids, MI 49503

31st Jan 2024

Re: EMBOJ-2023-114888R

Pseudomonas aeruginosa CtpA adopts a novel activation mechanism to initiate the proteolytic process

Dear Huilin and Andrew,

Thank you for submitting your revised manuscript to The EMBO Journal. With some delay due to the holidays at the turn of the years, we have now finally received the feedback of the three original referees, copied below for your information. As you will see, all referees consider the study significantly improved and key concerns adequately answered. Still, they raise a varying number of specific issues that should be addressed before publication, through presentational modifications and/or additional experiments (e.g. to obtain statistical significance). I am therefore inviting you to a final round of minor revision, and would be happy to discuss what could concretely be done, esp. for addressing the remaining concerns of referee 3.

When preparing your final revised version, please also make sure to take care of the following editorial points:

- To make the title somewhat more explicit, I would suggest including the qualifier "protease" next to CtpA, and in turn abbreviating "*Pseudomonas*" to "P." here.
- On the abstract page of the manuscript, please include 4-5 general keyword terms to enhance searchability.
- Please rename the Conflict of Interest section into "Disclosure and Competing Interests Statement", in accordance with our updated Guide to Authors (<https://www.embopress.org/competing-interests>)
- Please adjust the format of the reference list and of the in-text citations according to EMBO Journal format (alphabetical order, author name et al + year.../up to 10 author names in the reference list before et al / please refer to our Guide to Authors for additional information on EMBO J reference format). Also, please double-check each citation for completeness, as page number/locator information seems to still be missing for some of them.
- Please ensure that data listed in the Data Availability section becomes publicly accessible at this point, latest upon formal acceptance.
- In the Appendix PDF: Please correct the nomenclature in all instances (Toc, header, legends) to "Appendix Table S1/2/3..." or Appendix Figure S1/2/3..." i.e. adding the missing "S" before the number. The same should also be done for all in-text references to Appendix tables and figures. Furthermore, please also change the Appendix reference list from numbered to ordered-by-author according to our guidelines.
- Finally, please provide suggestions for a short 'blurb' text prefacing and summing up the conceptual aspect of the study in two sentences (max. 250 characters), followed by 3-5 one-sentence 'bullet points' with brief factual statements of key results of the paper; they will form the basis of an editor-written 'Synopsis' accompanying the online version of the article. Please also upload a synopsis image, which can be used as a "visual title" for the synopsis section of your paper. The image should be in PNG or JPG format, and please make sure that it remains in the modest dimensions of (exactly) 550 pixels wide and 300-600 pixels high.

I am therefore returning the manuscript to you for a final round of revision, hoping you will be able to address the remaining referee and editorial points in a straightforward manner. Please do not hesitate to contact me should you have any questions in this regard!

With kind regards,

Hartmut

Hartmut Vodermaier, PhD
Senior Editor, The EMBO Journal

*** PLEASE NOTE: All revised manuscripts are subject to initial checks for completeness and adherence to our formatting guidelines. Revisions may be returned to the authors and delayed in their editorial re-evaluation if they fail to comply to the following requirements (see also our Guide to Authors for further information):

9) Digital image enhancement is acceptable practice, as long as it accurately represents the original data and conforms to community standards. If a figure has been subjected to significant electronic manipulation, this must be clearly noted in the figure legend and/or the 'Materials and Methods' section. The editors reserve the right to request original versions of figures and the original images that were used to assemble the figure. Finally, we generally encourage uploading of numerical as well as gel/blot image source data; for details see: embopress.org/page/journal/14602075/authorguide#sourcedata

At EMBO Press, we ask authors to provide source data for the main manuscript figures. Our source data coordinator will contact you to discuss which figure panels we would need source data for and will also provide you with helpful tips on how to upload and organize the files.

Further information is available in our Guide For Authors:

In the interest of ensuring the conceptual advance provided by the work, we recommend submitting a revision within 3 months (30th Apr 2024). Please discuss the revision progress ahead of this time with the editor if you require more time to complete the revisions. Use the link below to submit your revision:

Link Not Available

Referee #1:

In this revised manuscript, the authors have largely addressed my concerns with additional experiments. My major points were the mode of activation of CtpA by LdcA and the role of cooperativity or substrate identity, whether the model of TPR being important was supported by the results, and the importance of the TRL (renamed TIL in the revision) loop for CtpA activity.

Based on the revised experiments they now show that CtpA exists as a dimer at physiological temperatures and LdcA aids in the assembly to a fully hexameric state and combined with ITC data address the question of cooperativity well. They note in the reviewer's responses that another substrate PA1048 showed the same effects as PA1198 with some of their preliminary work but that they focused on PA1198 throughout the paper, which addresses the selectivity question. It is worth included that initial result in the supplementary data. In addition, they remark that the ITC data "not fit well with a curve assuming three sequential binding sites" and they should show that fit in the supplemental data as well (this comment refers to fig 7a in the revision, which only shows the raw ITC traces but no fits).

They address the model of the TPR by noting the speculative nature of the TPR interaction in the description of their model, though they still use the language 'LbcA most likely recognizes a substrate via its TPR domain' (line 376). However, the de-emphasis of the role of the TPR domain is appreciated, particularly as it is not needed for their overall interpretation of the structure they are showing.

They have performed additional extensive experiments with mutations and truncations of the TIL loop, showing that activity is only minorly affected with removal of the TIL. This implies that the TIL does not have major role in CtpA activity, which the authors acknowledge. They also remark that the TIL loop functions primarily in stabilizing the hexamer, but there does not seem to be any biochemical results supporting that stabilization. The only minor point I would suggest here is that they simply state that the TIL loop has no real functional impact, as bringing up the possibility of structural stabilization raises the question of why this was not tested given all the other clear biophysical studies and is unnecessarily distracting from the core very interesting aspects of the paper. However, even if they leave this in, the revision has addressed all my major concerns.

Referee #2:

The manuscript by Hao-Chi Hsu et al. reports the cryo-EM structure of the CtpA protease in complex with its activator/adaptor LbcA. New data provided during revision demonstrate that 1) most CtpA exists predominately as a dimer at 30{degree sign} and primarily as hexamer at 4{degree sign}C, 2) binding LbcA to CtpA dimer causes CtpA hexamerization; 3) there is no cooperativity between the three LbcA binding sites of the CtpA hexamer. However, additional mutagenesis data are not conclusive (please see below).

Major concern:

The authors added new data (Fig 6g) showing the role of TIL in the cleavage of the model substrate PA1198. However, the conclusion regarding the effect of the point mutations (E385A, L388A) and TIL is based on the quantification of two experiments. The experiments should be conducted in triplicates and statistical analysis should be provided to make this conclusion.

Referee #3:

The authors have carefully addressed the different aspects of my comments with the first manuscript, and provide additional experiments to support their revision and proposed mechanism for the LbcA - CtpA interaction and CtpA activation.

The manuscript is much improved, though some points need additional revision before publication:

1. To address the question of possible cooperativity in LbcA-induced hexamerization (trimerization of dimers) and activation of CtpA, the authors provide ITC and SEC experiments - Fig. 7. In their response, the authors write: "LbcA binding leads to both activation of the CtpA dimer and rapidly conversion of the dimer into the hexamer. Because LbcA binding is prior to the CtpA hexamerization, the binding of individual LbcAs to the three vertices of CtpA hexamer is non-cooperative (revised Supplemental Figs. 9-11, new Fig. 7)."

I do not follow the logic in the argument that LbcA-induced CtpA hexamerization cannot be cooperative because LbcA binding would be prior to CtpA hexamerization. What data are the authors referring to state this chronology in binding? They start from dimeric CtpA, so the the first LbcA binding will be to dimer, but there is no data to claim that successive binding is to CtpA dimers or 6:1 CtpA:LbcA hexamers.

In fact, the ITC and SEC data shown in Fig. 7 suggest the opposite. The SEC profiles show that substoichiometric concentrations of LbcA (i.e. 6:1 and 6:2) already induce CtpA dimer trimerization. The high MW species being formed in the respective samples seem to be gradually shifting to higher hydrodynamic radius. The labelling in Fig. 7c seems to suggest these are all regarded as CtpA (6) : LbcA (3), but how do the authors explain the shortening elution time? This may reflect CtpA hexamers with 1, 2 or 3 LbcA bound, which would then suggest cooperativity in CtpA hexamerization. The authors describe that even in absence of LbcA, CtpA is in a (temperature-sensitive) dimer - hexamer equilibrium. It seems quite reasonable that if LbcA binding of a dimer stabilizes the hexamer conformation of the dimer, this would lead to cooperative shift to hexamerization (without informing whether it also activates the non LbcA-bound dimers in the CtpA hexamer). Binding of additional LbcA copies could further stabilize the hexamer, and/or result in activation of CtpA.

Small cryoEM or nsTEM collects of the different SEC eluates could resolve this ambiguity; or the authors could monitor stoichiometry of the eluates by SDS-PAGE.

Note that in Fig7c, the labelling is not unambiguous, two dotted lines go to "CtpA hexamer with 3 LbcA"; do the authors suggest a range? Please adjust the figure labelling.

2. The ITC data in Fig. 7a were fit with one set of binding sites. However, the enthalpy plot appears to show at least two transitions and a suboptimal fit with the one set binding site curve. Moreover, the fitted numbers do not appear to correspond with the shown curves, i.e. the curves appear to show an inflection at 0.6 and 1.2 molar ratio. How does this result in a fitted N of 3.4 ? Did the authors try global fitting?

3. In the author response letter, the authors provide results tracking the substrate binding and proteolytic activity of CtpA as induced by LbcA mutants lacking the TPR domain. I would agree that at this stage, the results are preliminary and may better be incorporated in a future manuscript reporting the structure of a substrate bound CtpA:LbcA complex.

Minor points:

- Ln 309-310: "we performed isothermal titration calorimetry (ITC) to measure the kinetics of LbcA binding to CtpA(S302A) at 25C." Do the authors mean the thermodynamics of binding?

Point-to-point response to reviewers' comments

Referee #1:

In this revised manuscript, the authors have largely addressed my concerns with additional experiments. My major points were the mode of activation of CtpA by LdcA and the role of cooperativity or substrate identity, whether the model of TPR being important was supported by the results, and the importance of the TRL (renamed TIL in the revision) loop for CtpA activity.

Based on the revised experiments they now show that CtpA exists as a dimer at physiological temperatures and LdcA aids in the assembly to a fully hexameric state and combined with ITC data address the question of cooperativity well. They note in the reviewer's responses that another substrate PA1048 showed the same effects as PA1198 with some of their preliminary work but that they focused on PA1198 throughout the paper, which addresses the selectivity question. It is worth included that initial result in the supplementary data. In addition, they remark that the ITC data "not fit well with a curve assuming three sequential binding sites" and they should show that fit in the supplemental data as well (this comment refers to fig 7a in the revision, which only shows the raw ITC traces but no fits).

Thank you for acknowledging we have largely addressed the previous major concerns. We have now also included experiments similar to Fig. 2i and Fig. 3h but using a different substrate (PA1048), and the new data is shown in **Appendix Figure S6**, which demonstrates that the proteolysis defects of the LbcA and CtpA mutants are not substrate specific (something we have never found in any of our stud. Per reviewer's request, we have also included the three-sites fitting in the revised **Appendix Figure S11**.

They address the model of the TPR by noting the speculative nature of the TPR interaction in the description of their model, though they still use the language 'LbcA most likely recognizes a substrate via its TPR domain' (line 376). However, the de-emphasis of the role of the TPR domain is appreciated, particularly as it is not needed for their overall interpretation of the structure they are showing.

Thank you for the suggestion which we agree. We have rephrased the sentence to "*LbcA recognizes the substrates and delivers them for degradation by attaching its N-terminal adaptor domain onto the NDR of a CtpA dimer*".

They have performed additional extensive experiments with mutations and truncations of the TIL loop, showing that activity is only minorly affected with removal of the TIL. This implies that the TIL does not have major role in CtpA activity, which the authors acknowledge. They also remark that the TIL loop functions primarily in stabilizing the hexamer, but there does not seem to be any biochemical results supporting that stabilization. The only minor point I would suggest here is that they simply state that the TIL loop has no real functional impact, as bringing up the possibility of structural stabilization raises the question of why this was not tested given all the other clear biophysical studies and is unnecessarily distracting from the core very interesting aspects of the paper. However, even if they leave this in, the revision has addressed all my major concerns.

We agree and have removed the mention of stabilization. We have revised the text to "*However, removing the TIL did not reduce CtpA protease activity significantly. As the TIL is specific for CtpA and is ordered only in the CtpA hexamer bound to three LbcA, it is likely that the loop's function is primarily related to the active CtpA hexamer. The TIL concatenates the six CtpA subunits, but the requirement of CtpA hexamerization remains to be investigated*".

Referee #2:

The manuscript by Hao-Chi Hsu et al. reports the cryo-EM structure of the CtpA protease in complex with its activator/adaptor LbcA. New data provided during revision demonstrate that 1) most CtpA exists predominately as a dimer at 30°C and primarily as hexamer at 4°C, 2) binding LbcA to CtpA dimer causes CtpA hexamerization; 3) there is no cooperativity between the three LbcA binding sites of the CtpA hexamer. However, additional mutagenesis data are not conclusive (please see below).

Major concern:

The authors added new data (Fig 6g) showing the role of TIL in the cleavage of the model substrate PA1198. However, the conclusion regarding the effect of the point mutations (E385A, L388A) and TIL is based on the quantification of two experiments. The experiments should be conducted in triplicates and statistical analysis should be provided to make this conclusion.

Thanks for raising this issue. We have now performed a new set of protease assays. The average and standard deviation from three measurements of each mutant are now shown in **Appendix Figure S10**, so that readers have all of the information needed to assess our conclusion.

Referee #3:

The authors have carefully addressed the different aspects of my comments with the first manuscript, and provide additional experiments to support their revision and proposed mechanism for the LbcA - CtpA interaction and CtpA activation.

The manuscript is much improved, though some points need additional revision before publication:

1. To address the question of possible cooperativity in LbcA-induced hexamerization (trimerization of dimers) and activation of CtpA, the authors provide ITC and SEC experiments - Fig. 7. In their response, the authors write: "LbcA binding leads to both activation of the CtpA dimer and rapidly conversion of the dimer into the hexamer. Because LbcA binding is prior to the CtpA hexamerization, the binding of individual LbcAs to the three vertices of CtpA hexamer is non-cooperative (revised Supplemental Figs. 9-11, new Fig. 7)."

I do not follow the logic in the argument that LbcA-induced CtpA hexamerization cannot be cooperative because LbcA binding would be prior to CtpA hexamerization. What data are the authors referring to state this chronology in binding? They start from dimeric CtpA, so the first LbcA binding will be to dimer, but there is no data to claim that successive binding is to CtpA dimers or 6:1 CtpA:LbcA hexamers.

In fact, the ITC and SEC data shown in Fig. 7 suggest the opposite. The SEC profiles show that substoichiometric concentrations of LbcA (i.e. 6:1 and 6:2) already induce CtpA dimer trimerization. The high MW species being formed in the respective samples seem to be gradually shifting to higher hydrodynamic radius. The labelling in Fig. 7c seems to suggest these are all regarded as CtpA (6) : LbcA (3), but how do the authors explain the shortening elution time? This may reflect CtpA hexamers with 1, 2 or 3 LbcA bound, which would then suggest cooperativity in CtpA hexamerization. The authors describe that even in absence of LbcA, CtpA is in a (temperature-sensitive) dimer - hexamer equilibrium. It seems quite reasonable that if LbcA binding of a dimer stabilizes the hexamer conformation of the dimer, this would lead to cooperative shift to hexamerization (without informing whether it also activates the non LbcA-bound dimers in the CtpA hexamer). Binding of additional LbcA copies could further stabilize the hexamer, and/or result in activation of CtpA.

Small cryoEM or nsTEM collects of the different SEC eluates could resolve this ambiguity; or the authors could monitor stoichiometry of the eluates by SDS-PAGE.

Thank you for raising this issue. Particle counting in electron micrographs can be subjected to variation in grid preparation condition and relies on the somewhat subjective particle picking process. Because of this, the EM based counting result can sometime be controversial. Therefore, we have opted to follow the reviewer's advice to perform an SDS-PAGE analysis of the assembled peaks. The new results show that all high MW species had the same molar ratio of 2:1. This data is now presented as a new panel (revised **Fig. 7d**). Because CtpA exists primarily as a dimer in solution (**Fig. 7a**), and LbcA binding to the CtpA dimer is a single-site binding event, which is further borne out by our ITC measurement (**Fig. 7b**). Therefore, there is perhaps no cooperativity issue at this initial binding event (i.e., CtpA hexamer is a minor species, and for the purpose of ITC curve fitting, we assume it does not yet exist at the point of LbcA binding).

We call out three peaks in the gel filtration profiles (peaks 1, 2, and 3; **Fig. 7c**). Peak 1 is the fully assembled CtpA₆LbcA₃. This peak is small, indicative of inefficient assembly in vitro with separately purified CtpA and LbcA, and this is supported by the fact that we had to resort to co-expression to produce enough of this complex for structural analysis (See **Methods**). Peak 3 is CtpA dimer only. The peak 2 is interesting, as it is gradually shifting, as noted by the reviewer. We suggest that the shift arises from addition (partial overlapping) of the varying amount (varying peak height) of intermediates such as (LbcA₁CtpA₂)x1, (LbcA₁CtpA₂)x2, and (LbcA₁CtpA₂)x3 (= CtpA₆LbcA₃ = hexamer complex).

The fact that we observe only 6:3 complex (CtpA₆LbcA₃), but not 6:1 or 6:2 complexes (CtpA₆:LbcA₁ or CtpA₆:LbcA₂) (**Fig. 7c-d**) suggests to us that an LbcA-bound CtpA dimer (LbcA₁CtpA₂) can only assemble with other LbcA-bound CtpA dimers (LbcA₁CtpA₂) but NOT with a CtpA dimer without LbcA binding (CtpA₂). We have clarified this in the revised text.

Note that in Fig7c, the labelling is not unambiguous, two dotted lines go to "CtpA hexamer with 3 LbcA"; do the authors suggest a range? Please adjust the figure labelling.

Sorry for the mislabel. We have revised to make the labels clearer.

2. The ITC data in Fig. 7a were fit with one set of binding sites. However, the enthalpy plot appears to show at least two transitions and a suboptimal fit with the one set binding site curve. Moreover, the fitted numbers do not appear to correspond with the shown curves, i.e. the curves appear to show an inflection at 0.6 and 1.2 molar ratio. How does this result in a fitted N of 3.4 ? Did the authors try global fitting?

We apologize for mistakenly mixing the dimer vs hexamer calculation in the original figure. We have revised to consistently use the dimer-based analysis in revised **Fig. 7b** and have now included the sequential three-site fitting in **Appendix Figure S11**.

3. In the author response letter, the authors provide results tracking the substrate binding and proteolytic activity of CtpA as induced by LbcA mutants lacking the TPR domain. I would agree that at this stage, the results are preliminary and may better be incorporated in a future manuscript reporting the structure of a substrate bound CtpA:LbcA complex.

Thank you for agreeing with us not to present the preliminary data in this manuscript.

Minor points:

- Ln 309-310: "we performed isothermal titration calorimetry (ITC) to measure the kinetics of LbcA binding to CtpA(S302A) at 25C." Do the authors mean the thermodynamics of binding?

Thank you for catching the error. We have revised: "*we performed isothermal titration calorimetry (ITC) to measure the thermodynamics of LbcA binding to CtpA(S302A) at 25 °C*".

Prof. Huilin Li
Van Andel Institute
Structural Biology
333 Bostwick Ave NE
Grand Rapids, MI 49503

26th Feb 2024

Re: EMBOJ-2023-114888R1

P. aeruginosa CtpA protease adopts a novel activation mechanism to initiate the proteolytic process

Dear Huilin and Andrew,

Thank you for submitting your final revised manuscript for our consideration. I am pleased to inform you that we have now accepted it for publication in The EMBO Journal.

With kind regards,

Hartmut
